



# Hydrological connectivity from glaciers to rivers in the Qinghai-Tibet Plateau: roles of suprapermafrost and subpermafrost groundwater

Rui Ma[1,2*], Ziyong Sun[1,2*], Yalu Hu[1], Qixin Chang[1], Shuo Wang[1], Wenle Xing[1], Mengyan Ge[1]

[1] Laboratory of Basin Hydrology and Wetland Eco-restoration, China University of Geosciences, Wuhan, 430074, China
[2] School of Environmental Studies, China University of Geosciences, Wuhan, 430074, China

*Correspondence to*: Rui Ma (rma@cug.edu.cn) and Ziyong Sun (ziyong.sun@cug.edu.cn )

**Abstract.** The roles of subsurface groundwater flow in the hydrological cycle within the alpine area characterized by permafrost and/or seasonal frost are poorly known. We studied the role of permafrost in controlling groundwater flow and the hydrological connections between glaciers in high mountain and river in the low plain with hydraulic head, temperature, geochemical, and isotopic data. The study area was a catchment in the headwater region of the Heihe River in the northeastern Qinghai-Tibet Plateau. The groundwater in the high mountains mainly occurs as suprapermafrost groundwater, and in the moraine and fluvio-glacial deposits on the planation surfaces of higher hills suprapermafrost, intrapermafrost, and subpermafrost groundwater co-occur. Glacier and snow-meltwater are transported from the high mountains to the plain through stream channels, slope surfaces, and supra- and subpermafrost aquifers. Groundwater in the Quaternary aquifer under the piedmont plain is recharged by the lateral inflow from permafrost groundwaters and the infiltration of streams, and is discharged as baseflow to the stream in the north. Groundwater maintained stream flow over the cold season and significantly contributed to the stream flow during the rainy season. $^3$H and $^{14}$C data indicated that the age of supra- and sub-permafrost groundwater, and groundwater in Quaternary aquifer of seasonal frost zone, ranges from 30-60 years. Two proposed mechanisms contribute to seasonal variation of the aquifer water-conduction capacity: (1) surface drainage through the stream channel during the high-flow period, and (2) subsurface drainage to an artesian aquifer confined by stream icing and seasonal frost during the cold season.

## 1. Introduction

The role of permafrost in groundwater flow systems is important in the hydrological cycles of cold regions (Walvoord et al., 2012). This is especially true for the mountain headwaters of large rivers. In these areas interactive processes between permafrost and groundwater influence water resource management, engineering construction, biogeochemical cycling, and downstream water supply and conservation (Cheng and Jin, 2013). Study of groundwater in permafrost areas has been prompted by the need for water supplies, problems associated with groundwater in mining, and construction of buildings, highways, railways, airfields and pipelines. The ice features of permafrost areas and geological mapping are also of great interest (Woo, 2012).





In permafrost-dominated watersheds, hydrogeological regimes are primarily controlled by the distribution of frozen ground and taliks, as well as the freeze-thaw cycle of aquifers (White et al., 2007). Freezing alters the intrinsic behavior of aquifers because ground ice occupies interstitial voids and reduces the permeability of the water storage matrix (Woo, 2012). The freeze-thaw process of the active layer affects the groundwater flow path and its interaction with surface water. Permafrost greatly affects base flows in the thawing season and groundwater and surface-water interactions in permafrost regions (e.g. Bense and Person, 2008; Carey and Quinton, 2005; Woo et al., 2008; Zhang et al., 2013). Groundwater-surface water interactions in Alaska are more commonly found in areas of discontinuous permafrost (e.g. Anderson et al., 2013; Minsley et al., 2012; Walvoord et al., 2012). Hydraulic connections are altered as a result of permafrost freeze and thaw (Carey and Woo, 2000). At high latitudes, permafrost distribution may affect lake density in addition to surface flow (Anderson et al., 2013). In areas of continuous permafrost, subpermafrost groundwater is often isolated from the surface, and there are unique mechanisms in thermokarst lake dynamics such as lateral expansion and breaching (Jones et al., 2011; Plug et al., 2008). Permafrost is now warming and thawing in many areas of Alaska (e.g. Anderson et al., 2013), and connections between permafrost degradation and local hydrologic changes have been established (e.g. O'Donnell et al., 2012; Yoshikawa and Hinzman, 2003).

Fundamental knowledge of groundwater systems in areas of permafrost is often lacking (Kane et al., 2013). Groundwater behavior in permafrost-dominated areas is understudied but will become more important as permafrost, an effective barrier to recharge, continues to degrade. Altering the proportion of groundwater to total discharge will shift the composition of biogeochemical exports (Walvoord and Striegl, 2007). Permafrost hydrogeology studies have been limited to research on groundwater chemistry and modeling, mostly in Alaska (USA), Canada, Siberia, Fennoscandia, and Antarctica (Bense et al., 2009; Carey and Quinton, 2005; Evans et al., 2015; Ge et al., 2011; Woo et al., 2008). Linkage between groundwater circulation and discharge has not been found in field studies but simulations have shown a possible connection between changes in climate, groundwater movement, and increases in the winter low flows of northern Eurasian and northwestern North American rivers (Smith et al., 1991; Walvoord and Striegl, 2007). The quantitative substantiation of linkage is challenging because hydrogeological; and permafrost information is lacking in remote areas and also due to the complexities of regional-scale permafrost-hydrology interactions. Lack of typical hydrogeological; information such as hydraulic head data, detailed hydrostratigraphy, and groundwater age data, impedes development of detailed models (Walvoord et al., 2012). Thus, an initial conceptual understanding of groundwater flow systems in permafrost regimes is essential for advanced numerical modeling to quantitatively analyze groundwater flow and its interaction with surface water.

Because of the limited infrastructure and short field seasons in remote areas, geochemical and isotope tracers in samples from baseflow discharge and springs have been used to study recharge conditions and flow paths. Stotler et al. (2009) investigated the role of permafrost in influencing deep flow system evolution, fluid movement and chemical evolution using hydrogeochemistry and H and O isotopes. Anderson et al. (2013) investigated the causes of lake area changes in Yukon Flats, a region of discontinuous permafrost in Alaska and found that about 5% of lake water comes from snowmelt and/or permafrost thaw with H and O isotopes. Utting et al. (2013) used stable isotopes ($\delta^{18}$O, $\delta$D and $\delta^{13}$C$_{DIC}$) and noble gases to





explore groundwater recharge and flow from permafrost watersheds in the western Arctic of Canada. Geochemical and isotopic data proved useful in delineating the groundwater system and identifying flow paths in the permafrost zone. However, the related research was limited to Canadian and Fennoscandian Shield groundwaters, Alaska, and other part of the USA.

To better understand the effects of permafrost on groundwater flow and interactions with surface water, we selected a representative catchment study site in the headwater region of the Heihe Basin. This area is covered by large areas of continuous and discontinuous permafrost and seasonal frost. The Heihe River is the 2nd largest inland river of China with a drainage area of ~150,000 km$^2$ (Figure 1(a)). It provides water for domestic use, agriculture, and industry in the Qinghai, Gansu and Inner Mongolia Provinces of northwestern China. The hydraulic head and temperature data obtained from newly drilled wells as well as geochemical and isotope information were combined and used to: 1) trace the recharge and flow paths of groundwater; and 2) investigate the control of permafrost and seasonal frost distribution and its freeze-thaw processes on groundwater dynamics and its interaction with surface water. Moraine and fluvio-glacial deposits are widely distributed on the planation surfaces of the higher hills in the headwater region of the Heihe River (Figure 1(a)). However, their significance for controlling groundwater recharge and flow has not been studied. This is the first report on the occurrence of subpermafrost and intrapermafrost groundwater in this region and their hydraulic connectivity with groundwater in the seasonal frost zone and river in the headwater regions of the Heihe River. Our research will increase understanding of planation surface areas that provide a major reservoir for the storage and flow of groundwater and rivers in permafrost regions.

## 2. Study area and background

### 2.1 General setting

The Hulugou catchment has a drainage area of 23.1 km$^2$ and is located within Qilian Mountain in the northern Qinghai-Tibet plateau, between 38°12'14"N and 38°16'23"N latitude and 99°50'37"E and 99°53'54"E longitude (Figure 1(b)). The Hulugou catchment elevation ranges from 2960 to 4820 m increasing from north to south. The slope ranges from 0° and 85°.

The catchment has a continental semi-arid climate characterized by warm, rainy summers and cold, dry winters. From the plain to the high mountains, the mean annual precipitation ranges from 400−600 mm, approximately 70% of which occurs during July–September (Figure 2). In the high mountains with elevations from ~3800 to 4800 m, most precipitation falls as snow. Snow may fall in summer, but it typically melts within one to three days. Evaporation ranges from 376 to 650 mm per year. The mean annual temperature is -3.9 °C, and the minimum and maximum temperatures are -25.2 °C and 25.8 °C, respectively. The daily precipitation and temperatures in the plain from 2014 through 2016 are shown in Figure 2.

The catchment geomorphology is composed of high mountains, erosion hills, and a piedmont sloping plain. High



mountains are located in the southern part of the catchment, and this area contains five alpine glaciers, two ice lakes, and a range of classic glacial features such as U-shaped valleys, cirques, horn peaks, arêtes, moraines and talus slopes. The five glaciers have a total area of 0.827 km$^2$ (Li et al., 2014). The erosion hills are in the northeast and northwest of the catchment, with the planation surfaces on the top of the higher hills (3400−3800 m) (Xu et al., 1989). The planation surfaces are

underlain by permafrost, with typical permafrost-related features such as thermokarst ponds, frost mounds, permafrost bogs, and permafrost plateaus. Cracks, terraces and landslides caused by active layer detachment slides are common on the upper slopes. The sloping plain is composed of several partially superimposed alluvial-pluvial fans. It is funnel-shaped, surrounded by the high mountains and hills, and having a narrow gorge at the base, which leads into the Heihe River (Figure 1(c)). There is a distinct break in the slope between the plain and the mountains. The plain dips slightly toward the Heihe River with 2−3

degree slopes.

The Heihe River is fed by the east tributary and west tributary in front of the narrow valley (Figure 1(b)). Both tributaries and their branches originate from the high mountains and are fed mainly by glacial and snow meltwater, ice lakes, and springs. From headwaters to the plain, they receive runoff from subcatchments which are derived from precipitation, and this increases the volume of water flow. The tributaries and their branches are all ephemeral streams. They are intermittently dry

throughout the October to May cold season. Only the main stream in the narrow gorge is perennial, though it is ice covered during winter.

## 2.2 Hydrogeology

Bedrock in the high mountains comprises lower Ordovician metamorphic rock and volcanic rock, including interbedded meta-sandstone and slate with local intermediate basic volcanic and green basal with local crystalline limestone (Xu et al.,

1989). Late Quaternary moraine deposits, derived primarily from these formations, are located at the front of glaciers within cirques (Figure 1(c)). The moraine is 5−30 m thick and consists of non-sorted, angular gravels and boulders. Scree deposits are also common in the high mountain area, and generally located at the base of steep rock slopes or valley walls.

Bedrock in the erosion hills is comprised of shales with limestone and sandstone (Xu et al., 1989). The slopes are generally covered with weathered residues of 0.5−3 m thickness but can also have local areas of exposed bedrock, talus

material, and silt deposits. The top of the higher hills, recognized as planation surfaces, are covered with middle and upper Pleistocene moraine and fluvio-glacial deposits from several meters to tens of meters thick (Cao, 1977). Thin mud deposits are also found here, especially in thermokarst ponds, permafrost bogs, and permafrost plateaus.

The surface geology in the piedmont sloping plain is primarily upper Pleistocene fluvio-glacial deposits (Xu et al., 1989), which are mainly composed of poorly-sorted, subangular, mud-bearing pebble gravels with erratic boulders. The underlying

strata are glacial moraine and fluvio-glacial deposits of the middle and lower Pleistocene series and conglomerates and sandstones from the Cretaceous (Xu et al., 1989). The Holocene alluvial-proluvial deposits are only found on the bottom of the narrow defile of the Hulugou stream gorge. Near the outlet of the Hulugou catchment, the upper Quaternary



alluvial-proluvial deposits occur on the first to third terraces of the Heihe River.

The groundwater flows correspond to the topography, with a flow trend from south to north. According to previous regional hydrogeological investigations (1:200,000) (Cao, 1977), permafrost in the head water regions of the Heihe River mainly occurs in areas exceeding 3600 m and the groundwater in permafrost region was conjectured to be suprapermafrost groundwater. Neither subpermafrost nor intrapermafrost groundwater has been reported. Our field investigation demonstrates that permafrost can be found at as low as 3500 m a.s.l. in shady slopes We found springs or seeps at the lower margin of the cirques containing moraine and scree deposits and the upper slopes of the hills with fluvio-glacial deposits on the top planation surfaces. Groundwater in the seasonal frost region primarily occurred in the mountain scree deposits, slope deposits of the hills, and fluvio-glacial deposits of the sloping piedmont plain.

## 3. Materials and methods

### 3.1 Field measurement

Four cluster wells, WW01, 02, 03 and 04, were installed in July and August, 2014 for groundwater monitoring and sampling (locations shown in Figure 1(b)). Each cluster well included 3−4 wells with different interval screen depths. The screened intervals were 5, 10, 15 and 25 m underground for cluster WW01, 5, 10, 20 and 30 m underground for clusters WW02 and WW03, and 1.5, 12 and 24.3 m for cluster WW04. No water was found in wells within the WW02 cluster. Cluster wells WW01, 02 and 03 were located in the piedmont sloping plain dominated by seasonal frost, at elevations of 3144, 3250 and 3297 m, respectively. Cluster WW04 was located in a planation surface dominated by thermokarst ponds, frost mounds, and permafrost bogs, at an elevation of 3501 m.

During installation of each cluster well, temperature loggers (HOBO U20-001-02 temperature logger; Onset, Bourne, MA, USA) were buried in sediments at depths of 0.5, 1, 1.5, 2, 3, 5, 10, 15 (20) and 25 (30) m below the ground surface to monitor the temperature profile with time. The ground temperature was recorded at an interval of 15 min. Both water table, available in the cluster wells, and river stage were measured using electronic pressure sensors (HOBO U20-001-02 water level logger; Onset, Bourne, MA, USA). The sensor for river water pressure measurement was installed in a stilling well to exclude waves and turbulence. Atmospheric pressure was measured simultaneously using a barometric pressure sensor (S-BPB-CM50; Onset, Bourne, MA, USA), so that differential pressure between water pressure and atmospheric pressure could be calculated and then converted to the water table. The data was recorded every 15 min to be consistent with ground temperature measurements.

Five weather stations have been maintained by Cold and Arid Regions Environmental and Engineering Research Institute, Chinese Academy of Sciences since 2004 within the Hulugou catchment. These stations collect air temperature, humidity, precipitation, and wind speed data at 30 min intervals (Chen et al., 2014). Data from the two stations on the sloping plain and the station near cluster WW04 (~200 m away; similar elevation) were collected in this study.




## 3.2 Water sampling and analysis

For ion and isotope analysis, groundwater samples were collected from the 12 wells between 2014 - and 2016 and stream water samples were collected from 12 sites that were approximately evenly distributed from upstream to downstream between 2011 and 2016. Sample sites are shown in Figure 1(b). Both types of samples were collected at 7 to 14 d intervals

during the thaw season from June to September, but less frequently during the cold season. They were collected 3-4 times in each of January and April. In addition to the 12 regularly sampled wells, groundwater was also occasionally sampled from 7 springs and 18 shallow wells less than 3 m in depth. Glacier meltwater was collected at 13 periglacial sites at elevations from 4261 to 4432 m between 2013 and 2015. Weekly precipitation (rainfall or/and snowmelt) was sampled from 3 sites that were distributed at about 200 m elevation intervals between 2012 and 2015.

Seven water sample subsets were collected from each site and filtered with 0.22 $\mu m$ membranes in the field into polythene bottles that were thoroughly pre-washed with deionized water. When groundwater was collected from wells, appropriate well purging was done using a peristaltic pump before sampling. At all sampling times, pH, electric conductivity (EC), temperature, and dissolved oxygen concentration were measured in the field using a portable Hatch Ec and pH meter (HACH HQ40d), and alkalinity was determined on the sampling day using the Gran titration method. Samples for cation and

minor element analysis were acidified with ultrapure $HNO_3$ to pH=2. All samples were wrapped with parafilm and stored at 4 ℃ before being transported to the laboratory for ion and isotope analysis.

All samples were analyzed for major ions, minor elements (Fe, Si and Sr), $^{18}O$ and $^2H$ isotopic compositions, and $^{13}C$ isotopic compositions of DIC. Thirteen groundwater and spring samples were analyzed for $^3H$ concentrations and 7 were analyzed for $^{14}C$ activity. Anions ($SO_4^{2-}$, $Cl^-$, $NO_3^-$) were determined using ion chromatography (IC; DX-120, Dionex, USA),

whereas cations ($Ca^{2+}$, $Mg^{2+}$, $K^+$, $Na^+$) and some minor elements (Si, Sr) were determined by inductively coupled plasma-atomic emission spectrometry (ICP-AES; IRIS INTRE II XSP) at the Laboratory of Basin Hydrology and Wetland Eco-restoration, China University of Geosciences (Wuhan) within 14 d after sampling. Ionic balance errors were < 5% for 84% of the samples and between 5.1%−8% for the remaining samples.

Isotopic compositions of $^{18}O$ and $^2H$ were analyzed using an ultra-high precision isotopic water analyzer (L2130-I, Picarro,

USA) at the Laboratory of Basin Hydrology and Wetland Eco-restoration, China University of Geosciences (Wuhan), and were expressed in $\delta$ per milliliter relative to the V-SMOW (Vienna Standard Mean Ocean Water), with precision of 0.025‰ and 0.1‰, respectively. The $^3H$ concentration was determined by the solid polymer electrolysis enrichment method with a tritium enrichment factor of 10 using an LSC-LB1 Liquid Scintillation Counter (Quantulus 1220™). The detection limit for the tritium measurement was approximately ±1 TU. The $^3H$ values were reported in tritium units (TU).

The $\delta^{13}C$ value of DIC in water samples was measured using a wavelength scanning cavity ring-down spectroscopy (WS-CRDS; G2131-I, Picarro, USA), and reported as per milliliter relative to V-PDB. The analytical precision for $\delta^{13}C_{(DIC)}$ was 0.1‰. For measuring $^{14}C$, water samples were treated first with 85% phosphoric acid and filtered to remove weathering carbonates. $CO_2$ was purified and collected with a cryotrap, and then reduced to graphite using the Zn/Fe method. Finally,



$^{14}$C activity was determined using an accelerator mass spectrometry (AMS; 3 MV, Tandetron) at the Xi'an AMS Center. $^{14}$C activity was reported as percent modern carbon (pmC) and the analytical precision was 2‰.

### 3.3 Sediment sampling and analysis

Sediment samples were collected at 30 cm to 100 cm intervals along the profile when drilling the deepest borehole within each cluster. The subset for stable isotopic analyses was placed in an 8-mL borosilicate glass vial sealed with a Teflon-lined screw cap and parafilm, and stored at -20 °C. After being transported to the laboratory, water was extracted from the sediment samples using cryogenic vacuum distillation technique and then measured for $\delta^{18}$O and $\delta^{2}$H (Smith et al., 1991; Sternberg et al., 1986). The measuring instrument, method and precision were the same as that noted above.

### 3.4 $^{14}$C age model

Along the groundwater flow path, the $^{14}$C gained in soils is often diluted by geochemical reactions such as carbonate dissolution, exchange with the aquifer matrix, and biochemical reactions. Therefore, when using the decay of $^{14}$C$_{DIC}$ as a measure of groundwater age, the dilution by non-atmospheric sources must first be corrected. Accounting for the dilution of $^{14}$C caused by geochemical reaction, groundwater age is calculated using the following decay equations (Clark and Fritz, 1997):

$$t = 8267 \times \ln\left(\frac{q \times {}^{14}C_0}{{}^{14}C}\right) \tag{1}$$

where $t$ is the groundwater age in years BP, $^{14}C$ is the measured $^{14}$C activity, $^{14}C_0$ is the modern $^{14}$C activity in the soil derived from DIC and $q$ is the dilution factor or fraction.

Several models have been proposed to obtain the dilution factor (e.g. Mook, 1980; Pearson and Hanshaw, 1970; Tamers, 1975; Vogel, 1970, 1967; Vogel and Ehhalt, 1963). In this study, a $\delta^{13}$C-mixing model modified by Clark and Fritz (1997) from the Pearson model (Pearson and Hanshaw, 1970) was applied to correct the $^{14}$C dilution by carbonate dissolution. This model is based on variations in $^{13}$C abundance, which differs significantly between the soil-derived DIC and carbonate minerals in the aquifer and is thus a good tracer of DIC evolution in groundwaters. Any process that adds, removes or exchanges carbon from the DIC pool and which thereby alters the $^{14}$C concentrations will also affect the $^{13}$C concentrations (Clark and Fritz, 1997). Therefore, the $q$ can be obtained from a $^{13}$C mass balance:

$$q = \left(\frac{\delta^{13}C_{DIC} - \delta^{13}C_{carb}}{\delta^{13}C_{rech} - \delta^{13}C_{carb}}\right) \tag{2}$$

where $\delta^{13}C_{DIC}$ is the measured $\delta^{13}$C in groundwater, $\delta^{13}C_{carb}$ is the $\delta^{13}$C of the calcite being dissolved (usually close to 0‰), and $\delta^{13}C_{rech}$ is the initial $\delta^{13}$C of DIC in the infiltrating groundwater. The $\delta^{13}C_{rech}$ was taken −18‰ as suggested by Han et al. (2011) for north China.



## 4. Results

### 4.1 Sediments

Well logs for cluster wells WW01−03 indicated that the sediments are mainly composed of sandy gravels which are highly permeable in the seasonal frost area (Figure 3). A silt clay layer with thickness between 3-6 m was found at all three sites and this may extend throughout the sloping plain. The underlying bedrock was not revealed by the deepest boreholes at the cluster wells WW01 and WW02, indicating that unconsolidated sediments are thicker than 25 m and 30 m at the two sites, respectively. At cluster well WW03, weathered sandstone was found at 22 m, indicating decreased thickness of unconsolidated sediments at the top of the alluvial-pluvial fans. These data suggest that alluvial-pluvial deposits might accumulate on a slightly sloping shallow saucer-shaped basin, being the thickest at the center of the plain and becoming thinner towards its edges (Figure 1(c)).

The sediments at WW04 consist of a top clay layer to the depth of 2 m, icy sandy gravel from 2−20 m, and ice-free sandy gravel at 20−25 m depth. There was a 0.2 m thick sandy clay layer at the depth of 12 m which was also ice free.

### 4.2 Groundwater depth

No liquid water was found in any well within cluster WW02 throughout the years, and in the 5 m and 10 m deep wells within cluster WW03 at low flow time in winter, and in the 12 m well within cluster WW04 at most times. Figure 4 shows variation of groundwater depth over time from 2014 to 2016 in the wells with groundwater. The groundwater depth in the 20 m and 30 m wells within cluster WW03 fluctuated between 13 m and 18 m below ground during the warm, rainy season from mid-June to late October. Conversely, the groundwater depth declined and was stable in the cold, rainless season. The water table in the 20 m well was always higher than that in the 25 m well, but the difference ranged from ~ 4 m in the cold season to less than 1 m in the warm season. From late June to late July, the water table in the 20 m well was similar to that in the 30 m well. The groundwater in 5 m, 10 m, 15 m and 25 m wells within cluster WW01 was shallow (4−6 m belowground) in the warm season, and dropped dramatically to 5−19 m below ground in the cold season. Although the water table differed greatly between the two seasons, it was relatively stable within either season. In contrast, the water table in 5 m well was comparatively stable throughout the whole year, ranging only from 4 m to 5 m in depth. This table was very close to, also followed the same seasonal variation trend with, the level of the adjacent Hulugou stream. Similarly, the water table in the shallower well was always higher than that in the deeper well within cluster WW01, but the difference was much smaller during the warm season. The water table depth ranged from 0 to 1.5 m below ground surface at the 1.5 m well within the active layer of permafrost zone at cluster WW04 during the warm season and the water froze in winter. The water table was observed in summer in the 24.3 m well at cluster WW04 and water depth varied between 20.3 and 23.5 m. It was difficult to be monitored automatically in winter because the water was too shallow to cover the pressure sensor.



### 4.3 Ground temperature

The profiles in Figure 5 show the inter-monthly variation of ground temperature over a year from September 2014 to August 2015 at each cluster. The upper part of the profiles was obviously influenced by seasonal heating and cooling from the land surface, showing significant seasonal changes in temperature. The temperature decreased at a gradually reduced rate

with depth in the warm season and this was reversed in the cold season. This is presented as right-concave profile and left-concave profile in temperature vs. depth graph, respectively. Two types of profiles converged at a critical depth where seasonal variation in temperature disappeared. The critical depth was about 7.5 m, 10 m and 12 m below the ground surface at the clusters WW03, 02 and 01, respectively, much deeper than that (only ~ 2 m) at cluster WW04.

However, a slight dynamic variation of temperature could still be observed below the critical depth at clusters WW01 and

WW03. It is probably due to the groundwater recharge or discharge processes, which is supported by comparison with the results at clusters WW02 and WW04. This dynamic variation was not found at depths between 10 m and 30 m at WW02 where groundwater depth exceeded 30 m, nor at depths between 2 m and 20 m at WW04 where temperature remained almost constant around 0 $^{o}$C and thus groundwater was frozen all year at these depths. A slight seasonal variation in temperature was observed below 20 m at cluster WW04. The pattern of the variation was similar to that in the upper part of

the profiles, i.e., increased temperatures in summer and decreased temperatures in winter.

The ground temperature profiles in the warm season did not intersect with the 0 $^{o}$C isotherm at the clusters WW01−03 (Figure 5), confirming that the three clusters are in a seasonal frost region. The seasonal frozen depth was about 7.5 and 10 m at clusters WW02 and WW03, shallower than that (12 m) at cluster WW01. The active layer was 2 m thick at cluster WW04.

### 4.4 Hydrogeochemistry

Chemical compositions are listed in Table 2. The lowest values in river waters from east tributary were located within the periglacial and permafrost zone, and increased in the waters from east and west tributaries located within the seasonal frost zone. The values further increased in the river water at the catchment outlet. On a seasonal basis, river water concentrations had larger geochemical variations compared to groundwater. Except for $NO_3^-$ and $SO_4^{2-}$, other major ions and minor elements (Si, Sr) as well as EC and TDS concentrations were similar between river water at the catchment outlet and

groundwater in winter (Table 2 and Figure 6). The spring waters exhibited minor changes of geochemistry between the seasons.

The groundwaters sampled from the seasonal frost region of the sloping plain (clusters WW01 and 03) were $HCO_3 \cdot SO_4$-Ca·Mg and $HCO_3 \cdot SO_4$-Mg·Ca types, being neutral to slightly alkaline with pH between 7.64 and 8.74. By comparison, samples at cluster WW01 had higher levels of major ions, Si and Sr, and TDS concentrations than those at

cluster WW03 (Table 1 and Figure 6 and 7). The $Ca^{2+}$, $NO_3^-$, $SO_4^{2-}$ and TDS concentrations in groundwater aquifers at relatively shallow depths (< 20 m) were generally higher than levels in deeper parts of the aquifer (> 25 m) at WW01 and 03 (Table 1 and Figure 7), whereas $Na^+$ and $K^+$ were higher in deeper parts of the aquifer. The other parameters ($Mg^{2+}$, $Cl^-$,





HCO$_3^-$, Si and Sr concentrations) were similar at different depths within the clusters WW01 or WW03.

The chemical type was HCO$_3$-Ca for suprapermafrost groundwater (1.5 m well at cluster WW04), HCO$_3$·SO$_4$-Mg·Ca for subpermafrost (24.3 m well at cluster WW04) and HCO$_3$-Ca·Na·Mg for intrapermafrost groundwater (12 m well at cluster WW04). Among the stream water samples, thermokarst water, and groundwater in three wells, the groundwater in the 12 m well (intrapermafrost groundwater) had the highest Ca$^{2+}$, Mg$^{2+}$, K$^+$, Na$^+$, Si, Sr, Cl$^-$, HCO$_3^-$ and TDS concentrations and the lowest SO$_4^{2-}$ and NO$_3^-$ concentrations. The groundwater in the 24.3 m well had the higher K$^+$, Na$^+$, Mg$^{2+}$, Sr, Cl$^-$, SO$_4^{2-}$ and NO$_3^-$ concentrations but lower Ca$^{2+}$ and HCO$_3^-$ concentrations than in the 1.5 m well. The supra- and subpermafrost groundwaters had similar TDS values and chemical compositions. The TDS and major ions concentrations except for Na$^+$ were lower than those from the seasonal frost region. The relative concentrations of Mg$^{2+}$ and SO$_4^{2-}$ showed little difference associated with depth.

### 4.5 Stable isotopes

A local meteoric water line (LMWL) drawn through the H and O isotopic composition of precipitation at the study area is $\delta^2$H= 8.5 $\delta^{18}$O+ 22.6 ($r^2$=0.9886; n=120) (Figure 8). This is similar to the line ($\delta^2$H= 8.3 $\delta^{18}$O+ 17.1) reported by Tong et al. (2016) at a weather station near the Hulugou streamcatchment outlet. The $\delta^{18}$O of glacier meltwater samples was between −10‰ and −7.6‰, respectively, while the $\delta^2$H was between −60‰ and −35‰, respectively. The $\delta^{18}$O and $\delta^2$H of river waters ranged between −12.3‰ and −6.7‰ and between −88.5‰ and −31.6‰, respectively (Figures 8 and 9). Most samples exhibited isotopic values that overlapped with the samples of groundwater from the seasonal frost zone (Figure 8).

The $\delta^2$H and $\delta^{18}$O of water or/and ice extracted from sediment cores at cluster WW04 were relatively positive at shallow depths (<5 m belowground), with values −10‰ ~ −50‰ for $\delta^2$H and −2‰ ~ −8‰ for $\delta^{18}$O (Figure 9). All samples from the depths fell below the LMWL, and could be statistically defined by the regression line: $\delta^2$H =6.17·$\delta^{18}$O + 2.99 ($r^2$=0.98) (Figure 8). The 6.17 slope of this line is less than that of the LMWL, indicating that the water/ice experienced evaporation (Clark and Fritz, 1997). In contrast, the $\delta^2$H and $\delta^{18}$O of water/ice extracted from sediment cores at depths between 5 m and 20 m were relatively negative, with average values of −50‰ for $\delta^2$H and −9.5‰ for $\delta^{18}$O (Figure 9), which were values similar to those of glacier meltwater. Most of the samples at the depths fell on the LMWL in the $\delta^2$H vs. $\delta^{18}$O plot, indicating a precipitation origin without significant evaporation. Below 20 m, however, the $\delta^2$H and $\delta^{18}$O values of water/ice extracted from cores were greater and fell on a line with a slope of 5.1. Groundwater in the 1.5 m well of cluster WW04 had the most positive $\delta^2$H and $\delta^{18}$O values and fell below the LMWL in the $\delta^2$H vs. $\delta^{18}$O plot (Figure 8) (Clark and Fritz, 1997). The groundwater samples collected from the 12 m well showed the most negative $\delta^2$H and $\delta^{18}$O values and were similar to the glacier meltwater samples in the $\delta^2$H vs. $\delta^{18}$O plot. The samples from 24.3 m well had intermediate $\delta^2$H and $\delta^{18}$O values between glacier meltwaters and the groundwaters from 1.5 m well in the $\delta^2$H vs. $\delta^{18}$O plot.

In the $\delta^2$H vs. $\delta^{18}$O plots, the groundwater samples collected from clusters WW01 and WW03 fell along the LMWL and were similar to the glacier meltwater samples and mountain precipitation samples. They were significantly depleted in $^2$H





and $^{18}$O compared to rainfall occurring on the plain. These groundwater samples had similar isotopic values and locations in the plot to the stream samples (Figure 8). By comparison, the samples showed more negative $\delta^2$H and $\delta^{18}$O values at cluster WW01 than at cluster WW03. A general depletion of $\delta^2$H and $\delta^{18}$O at depth of groundwater was observed at both clusters (Figure 10). However, a discrepancy or reversal of this general depletion happened frequently because the groundwater

showed significant variation in $\delta^2$H and $\delta^{18}$O and the variation differed at different depths (Figure 10).

### 4.6 Radioactive isotopes and groundwater age

The $^3$H concentrations were 15.11 TU in the groundwater sample at cluster WW04, between 16.20 to 24.18 TU in the water at clusters WW01 and WW03, and between 13.61 to 43.59 TU in the springs of the sloping plain (Table 2). Except for one spring sample (QW05), the $^3$H concentrations of all samples were < 30 TU, indicating that the groundwater was derived

from recent precipitation and some "bomb"  related $^3$H is possibly presented (Table 2) (Zhai et al., 2013). Along with flow path, $^{13}$C increased from the permafrost zone with values between -13.6 and -16.77 ‰ to higher locations of seasonal frost zone with values around -8.79‰, and further to lower locations of the seasonal frost zone with values around -5.09‰ (Table 2). The $^{14}$C activity in groundwater varied from 35.51 to 96.34 pmC (Table 2). Groundwater samples from cluster WW04 had much higher $^{14}$C values than values in groundwater and spring samples from the sloping plain, showing a general

increasing trend from the permafrost zone to higher locations of the seasonal frost zone, and further to the lower elevation groundwaters. Except for the 24.3 m well within cluster WW04, the corrected $^{14}$C ages of all samples were negative, indicating that they were derived from modern precipitation (Clark and Fritz, 1997). The sample from the 24.3 m well within cluster WW04, with the $\delta^{13}$C of −16.77‰, had a relatively old corrected $^{14}$C age of 1627 yr.

## 5. Discussion

### 5.1 Exchange and pathways of groundwater in the permafrost region

#### 5.1.1 Suprapermafrost groundwater

Though the $\delta^2$H and $\delta^{18}$O values indicate that suprapermafrost groundwater had experienced strong evaporation (Figure 8), it was still a HCO$_3$-Ca type in hydrogeochemistry, with low concentrations of TDS, Cl$^-$ and Na$^+$. This suggests that the suprapermafrost groundwater has a short flow path or rapid flow resulting in a relatively short residence time and relatively

weak water-rock interaction. This is supported by the highest $^{14}$C activity in the suprapermafrost groundwater among all samples, which was 96.34 pmC and close to the atmospheric value (Clark and Fritz, 1997), and a 15.11 TU $^3$H concentration, which is an indicator of modern water (Zhai et al., 2013). These data suggest that suprapermafrost groundwater is mainly recharged from recent local precipitation via vertical seepage. The widespread thermokarst ponds and organic cover with high porosity favor water entry into the suprapermafrost reservoir. This result was further confirmed by stable isotopic data,





which showed that the regression line through suprapermafrost groundwater samples intersects the LMWL near the precipitation samples. Given that cluster WW04 is located on the lowest of three ladder-like terraces, the suprapermafrost reservoir may also be recharged by lateral flow from a higher terrace. As indicated by the $\delta^2$H vs. $\delta^{18}$O plot, the lateral recharge should mainly be from the suprapermafrost aquifer located on a higher terrace.

The terrace on which cluster WW04 was located adjoins two opposite hill slopes created by streams cutting to the west and east, respectively, and a hill slope connecting to the north plain (Figure 1(c)). At the shoulder of the three slopes, the moraine and fluvio-glacial sediments arch over the slope, become thinner, and finally end at the upper slope. Thus, except for the portion that is discharged as evapotranspiration, much of the suprapermafrost groundwater flows to the adjacent slopes. Because these slopes are covered by weathered residues only 0-3 m thick, and with many local areas of exposed

bedrock and silt deposits (Xu et al., 1989), the suprapermafrost groundwater is mainly discharged directly into streams as baseflow, or onto the surface as seeps and springs and from there into streams. This explains why many springs and seeps are found on the upper slopes of the hills whose upper planation surfaces are covered with moraine and fluvio-glacial sediments. It is also a major reason why streams increase progressively in volume from headwaters to the sloping plain. Where weathered residues are continuous along the slope and have a coarse grain size, the suprapermafrost groundwater can be

discharged into these residues, then flow through them to the talus fan at the base of the hill and, finally, drain into the aquifers on the sloping plain. Our study demonstrated that another discharge way of suprapermafrost groundwater is leakage to the subpermafrost aquifer through sinkholes created by thawing and collapse of the permafrost (Figure 11).

        The amount of suprapermafrost groundwater in the active layer varies seasonally. It is recharged during the warm season because glacier melting and precipitation are mainly concentrated during this period. Meanwhile, the active layer undergoes

thawing. Recharge is limited in the cold season because recharge sources are frozen and active layer freezing obstructs infiltration (Woo, 2012). The discharge of suprapermafrost groundwater shows a corresponding seasonal cycle. An examination of groundwater depth and temperature data indicates that storage of suprapermafrost groundwater also varies significantly throughout the warm seasons. This is not only a result of variation in the thawed depth of the active layer, but is also related to the frequent conversion of recharge-discharge interrelationship. During late spring when the active layer is

beginning to thaw and the storage capacity of suprapermafrost reservoir is small, the water table is close to the surface and recharge is limited. In the summer, the groundwater can rise further and move over the land surface to support bogs and thermokarst ponds. At the same time, the seasonal thaw moves downward and the storage capacity of the suprapermafrost reservoir is increased. This is because the level of recharge is intensive and exceeds the discharge capacity of the aquifer. In October, the water table began to decrease and dropped to 1.2 m belowground by December. The surface water retreated to

the subsurface, leading to drying of the bogs and thermokarst ponds. The water table decline was caused by a reverse of the recharge-discharge interrelationships between surface water and groundwater due to the existence of permafrost. By late October, glaciers that are a major recharge source of suprapermafrost groundwater were frozen. Another major water source, local precipitation, was minimal. But discharge passages, located on hill slopes at relatively lower altitudes, remained unfrozen. Consequently, the discharge of suprapermafrost groundwater, exceeded the recharge during this period, which





resulted in the drainage of suprapermafrost groundwater, decline of the water table, and drying of bogs and ponds.

### 5.1.2 Subpermafrost groundwater

Our results demonstrate that subpermafrost groundwater has patterns similar to suprapermafrost groundwater regarding stable and radioactive isotopes and hydrogeochemistry or temporal variations of groundwater temperature. This suggests that the subpermafrost groundwater on the planation surface is strongly linked to surface hydrological processes. The ground temperature in the subpermafrost aquifer exhibits a slight seasonal variation consistent with air temperature. It increases in summer and decreases in winter. These results indicate the existence of an efficient passage allowing water to flow through permafrost from superficial water pools such as suprapermafrost aquifer, streams, or thermokarst ponds to the subpermafrost aquifer. An opposite seasonal variation in water was observed in subpermafrost groundwater compared to suprapermafrost water, rising in winter and declining in summer (Figure 4). This indicates suprapermafrost groundwater discharge to subpermafrost via passages such as sinkholes. There are widely distributed sinkholes resulting from thawing and collapse of permafrost that could serve as passages (Figure 11(a)), through which the suprapermafrost groundwater and thermokarst pond water rapidly recharge the subpermafrost groundwater. This is confirmed by the depleted $^{13}$C and relatively high DOC concentrations in subpermafrost groundwater.

However, the weaker evaporation and stronger water-rock interaction for subpermafrost groundwater, inferred by more depleted $\delta^2$H and $\delta^{18}$O compositions and larger TDS and major ion concentrations compared to those in suprapermafrost groundwater, suggest a second recharge source. This source could occur in a colder environment and thus be depleted in isotope composition and take a longer flow path and residence time in the recharging process. On the $\delta^2$H vs. $\delta^{18}$O plot, the subpermafrost groundwater falls between meltwater samples and suprapermafrost groundwater samples, suggesting that this second recharge source is glacier and snow meltwater. The terrace on which cluster WW04 is located adjoins two higher terraces that are composed of thick moraine and fluvio-glacial deposits. Further south are moraine sediments in cirques and a glacier. The thick unconsolidated sediment, consisting of highly permeable boulders and gravels, is continuous from the front of glacier to the lowest terrace on the top of the hill, and existence of a continuous, slightly sloping subpermafrost pore aquifer is expected. Thus, glacier meltwater recharging of the subpermafrost aquifer occurs mainly at localized water bodies such as glacier-fed headwater streams and lakes on the moraines, where surface water percolates through thawed stream bank and lake bed into the gravel and boulder deposits, then down into the subpermafrost aquifer and, finally, to the aquifer on the lowest terrace.

Groundwater depth data show that the subpermafrost groundwater table is always below the bottom of overlying permafrost. The well log also records a relatively dry sediment layer at depths between 12 m and 12.5 m. The occurrence of this kind of groundwater that is in the confining aquifer but without additional pressure is an indicator of poor recharge or/and good discharge of groundwater. As the hydrogeological setting is relatively favorable for the recharge of subpermafrost groundwater in summer, it must have a comparable discharge capacity. This capacity is related to aquifer



permeability. Our data on sediments and ground temperatures show that the permafrost on the planation surface is thin, and thus the subpermafrost groundwater is mainly stored in unconsolidated material that is composed of gravels and pebbles with high permeability. These would permit fast flow and thus facilitate groundwater discharge. In addition, the interface between unconsolidated sediments and underlying bedrock may also serve as an efficient passage for subpermafrost groundwater

discharge (Woo, 2012).

There is a distinct break in sediment composition and thickness between the planation surface and the adjacent three hill slopes. With the thinning of the moraine and fluvio-glacial sediments, the subpermafrost pore aquifer disappears over the impermeable bedrock or at the thin residues of the upper slopes. Thus, like the suprapermafrost groundwater, the subpermafrost groundwater is mainly discharged directly into streams as baseflow, or onto the surface as seeps and springs at

the upper portions of the hill slopes and then into streams. This is probably why several ground and spring icings can be found on the slopes during cold seasons (Figure 11(b)).

Subpermafrost groundwater is also recharged mostly in the warm season, whereas recharge is limited in the cold season because the sources are frozen. A similar pattern was seen for the discharge of subpermafrost groundwater. However, the starting time of discharge is earlier than the recharge time whereas the end times are reversed due to the altitude difference

between recharge sources and discharge exports. This would decrease subpermafrost groundwater storage in the early autumn and later spring and help explain why the subpermafrost groundwater table declines significantly in winter.

### 5.1.3 Intrapermafrost groundwater

Monitoring data of the 12 m well within cluster WW04 demonstrates the occurrence of intrapermafrost groundwater in a talik. However, the ground temperatures at the depths are similar to those of the neighboring upper and lower permafrost,

being ~ 0 °C throughout the year. Thus, the localized presence of this unfrozen cold groundwater is related to its high mineralization (~ 1059 mg/L in TDS). Hydrochemical and isotopic data further indicate that it is a closed talik. The EC, TDS and concentrations of major cations, minor elements (Si and Sr), $HCO_3^-$ and $Cl^-$ in intrapermafrost groundwater are much higher than in subpermafrost and suprapermafrost groundwater and thermokarst pond water, excluding the mixture of these open water sources. On the other hand, the intrapermafrost groundwater is $^2H$ and $^{18}O$ depleted and falls near the LMWL on

the $\delta^2H$ vs. $\delta^{18}O$ plot, indicating a modern meteoric water origin without significant evaporation. These results suggest that the higher TDS and ions in intrapermafrost groundwater are a result of long-term water-rock interactions in a closed environment. The well logs showed that this talik is rich in organic matter. Given the very low $SO_4^{2-}$ relative concentration (4.1 mg/L) and much higher $HCO_3^-$ concentration (833.6 mg/L) in the intrapermafrost groundwater, sulfurization may have occurred in the reservoir (Domenico and Schwartz, 1998). This provides additional evidence that this hydrochemical talik is

closed and possesses strong reducibility. The data also suggest that the intrapermafrost groundwater has a poor hydraulic connection with suprapermafrost and subpermafrost groundwater.



### 5.2 Exchange and pathways of groundwater in seasonal frost region

### 5.2.1 Groundwater at the top of the piedmont sloping plain

The water table in the 20 m well was always higher than that in the 30 m well within cluster WW03 and the table in both wells fluctuated greatly in response to heavy rainfall events during the warm season. This indicates that the groundwater at the top of the piedmont sloping plain is recharged mainly from rainfall or stream infiltration. However, this is not supported by the $\delta^2$H and $\delta^{18}$O values of groundwater, which were relatively constant and showed little response to rainfall or stream discharge pulses. The groundwater depth at cluster WW02 was > 30 m and we can use its ground temperature profiles as references that are not affected by groundwater flow. In comparison, the upper part (< 6 m in depth) of all profiles at cluster WW03 had a similar pattern, excluding the possibility of rainfall or stream infiltration which would have shifted the profile to the right on the depth vs. temperature plot during the warm season. Conversely, temperature profiles shifted to the left at 6−16 m depths in September and October 2014 and August 2015, and at 6-28 m depths in July 2015, indicating the presence of colder lateral inflows. The groundwater is significantly depleted in $^2$H and $^{18}$O compared with rainfall occurring in the plain, indicating its recharge sources were generated in a colder environment.

The combined groundwater level, temperature, and hydrogeochemical and isotopic data suggested that three sources may contribute to lateral inflows to the groundwater at the top of the plain. Two sources may be suprapermafrost and subpermafrost groundwater, which discharged onto hill slopes as seeps and springs and flowed down the slopes as surface runoff or discharged into and flowed through the weathered slope residues as subsurface runoff. The water then flowed into the talus fans at the base of the hill, and finally moved as a lateral flow into the aquifer at the top of the sloping plain. The third source is the suprapermafrost groundwater and surface runoff generated in the bedrock mountains which are connected to the south top of the piedmont sloping plain. In the bedrock outcrop area, only suprapermafrost groundwater occurs within the surface fissures and weathered zones and the amounts are limited (Cao, 1977). Bedrock areas were assumed to be key areas of surface runoff due to their steep slopes and low permeability (Chen et al., 2014). Much of this shallow subsurface and surface runoff flows into streams while much runoff flows through talus fans at the base of mountains and finally into the aquifer at the top of the sloping plain. The limited storage and rapid flow of this recharge source results in a significant water table response, at the top of the plain, to heavy rainfall events. The mixture of runoff generated at different altitudes minimizes the fluctuation of groundwater in $\delta^2$H and $\delta^{18}$O.

Ground temperature and water table results demonstrate that the lateral flow into the plain occurs mainly during rainy season and when the active layer thaws in the mountains. Talus consists of gravel and boulders that are more permeable than mud-bearing pebble gravels in the plain (Xu et al., 1989) and the lateral flow from the mountains may accumulate at the top of the plain. Thus, the water table there rises and the phreatic surface between mountain base talus and top of the plain becomes minor. As a result, the aquifer at the top of the plain is also dominated by lateral flow that has a small vertical component. This means little difference in hydraulic head between shallow and deep groundwater. It would explain the



distinct peaks in the water table level several days after heavy rainfall events and small difference in groundwater hydraulic head between the 20 m and 30 m wells during the rainy season. Over the cold season from November to June, the lateral inflow decreased and finally ceased completely. The groundwater stored during the rainy season was released slowly to the base of the plain. As a consequence, the water table at the top of the plain declined dramatically and thus the phreatic surface

between mountain base talus and the top of the plain became steeper. Therefore, the groundwater flow has a larger vertical downward component, which explains why the difference in water head between the 20 m and 30 m wells within cluster WW03 increased in the cold season. The lateral recharge from subpermafrost groundwater was probably continuous to mid-winter, as inferred by the slight shift of ground temperature profiles to the right in 15−28 m depth from October to December 2014. A depleted trend in $^2$H and $^{18}$O of groundwater during the cold season also provides evidence for this. The

change of the water table in cluster WW03 from falling to rising in April 2015 may mark the beginning of lateral recharge in a new annual cycle. This is consistent in time with the thaw of the active layer indicated by ground temperature results at cluster WW04.

### 5.2.2 Groundwater at the base of the piedmont sloping plain

Compared to the cluster WW03, the water table fluctuation at cluster WW01 was more gradual, representing typical

characteristics of groundwater in the discharge area. The groundwater had more negative $\delta^2$H and $\delta^{18}$O values at cluster WW01 than at cluster WW03, although groundwater was flowing from cluster WW03 to WW01 and isotopic enrichment of groundwater is expected along the flow path (Clark and Fritz, 1997). This means that an isotopically depleted water source must have recharged the groundwater when it flowed through the plain. The recharge by local rainfall that is isotopically enriched relative to groundwater can be excluded (Yang et al., 2012), so this additional recharge may have been from streams.

Streams are fed mainly by isotopically depleted glacier meltwater runoff and precipitation runoff in the high mountains and the water percolates down into the coarse-textured aquifer when flowing through the plain. This result is also supported by recent research on water balance in the plain, which reported that the thick vadose zone and high transpiration prevented precipitation from entering the aquifer (Chen et al., 2014). The $\delta^2$H and $\delta^{18}$O values of groundwater at cluster WW01, which were intermediate between values of groundwater samples at cluster WW03 and stream water samples, also support this

explanation (Figure 8 and 10). The $\delta^2$H and $\delta^{18}$O values of groundwater at cluster WW01 are closer to those of stream water during high-flow periods, indicating a larger contribution to the aquifer from stream leakage. The ground temperature profiles at cluster WW01, which shifted to the right below 12 m depth in September and October 2014 and July and August 2015, indicate that river leakage mainly occurred upstream in summer and then flowed toward the base of the plain as lateral inflows. The recharge of this "new" water source also explains the very young age of groundwater at cluster WW01 inferred

by $^3$H concentration and the TDS value similar to that in groundwater at cluster WW03.

As previously described, the piedmont sloping plain is funnel-shaped, with only a narrow gorge at the base leading to the Heihe River. The feeder streams converge into the main Hulugou stream in front of the gorge, which then is contained within





the gorge. Because the gorge is between hills that are composed of less permeable shales and sandstones and unconsolidated deposits are only found on the bottom of the gorge, groundwater from the open plain also converges in front of the gorge with the narrowing of flow cross section, and then is discharged mainly as baseflow along the main stream within the gorge or as springs at the base of the hill. This means that the groundwater in front of the joint of open plain and gorge is blocked and pushed upward similar to backwater caused by subsurface damming (McClymont et al., 2010). This is similar to the "fill and spill" mechanism in hillslope hydrology (Spence and Woo, 2003; Tromp-van Meerveld and McDonnell, 2006) and helps explain why the water table at cluster WW01, which is located slightly above the junction of open plain and gorge, was relatively high and stable in both rainy and dry seasons. Small differences in water head between the 5 m, 10 m, 15 m and 20 m wells during warm seasons indicates that the aquifer at the base of the plain is also dominated by lateral flow which has a small vertical component. Given that the groundwater flow is completely consistent in the horizontal direction with the stream flow within the gorge, the shallow groundwater should be mainly discharged into the upper portions of the main Hulugou stream, while the deep groundwater is discharged into the lower portions of the stream.

During the cold season, groundwater is still discharged mainly as baseflow, but the situation is complicated by development of stream icing and seasonal frost. Our data shows that all tributaries were dry throughout the October to May cold season and river icing was only found in the main Hulugou stream channel within the gorge (Figure 11(b)). Icing was initially formed in the upper channel in early winter, followed by continued thickening and downstream expansion of the icing in winter and early spring. At the same time, icing was also formed at the spring near the basin-hill border. Field investigation in late January 2015 showed that the upper reaches of the stream channel were completely filled with ice and no water was flowing under the ice. The streambed was probably also frozen, blocking groundwater discharge into the stream and also exerting hydrostatic pressure. Although there is no ground temperature data in the gorge, we can deduce from the data at cluster WW01 that the maximum depth of the seasonal frost should be >3 m. Considering that the main stream is sustained completely by baseflow in winter, the groundwater depth along stream channels (bottom of the gorge) should be shallow and probably < 1 m. Thus, the impermeable seasonal freezing would reach the water table rapidly in early winter and also exert pressure on groundwater. The unconsolidated sediment at the bottom of the gorge would then become a confined aquifer in winter. When groundwater flows from the phreatic aquifer in the open plain to this tilted confined aquifer, it would have a larger vertical downward component. This would explain why the difference in the water head between the 5 m, 10 m, 15 m and 20 m wells within cluster WW01 became larger in the cold season. At the lower reaches of the stream channel, though water was still flowing under ice, the increased icing constricted the channel cross section and exerted hydrostatic pressure on stream water, significantly reducing groundwater discharge into the channel (Kane, 1981). This may be another reason why the water table at the base of the open plain was relatively high and stable, even in the winter. The great difference in $\delta^2$H and $\delta^{18}$O between the groundwater at cluster WW01 and the stream water, which should be derived from the same source because the stream was sustained completely by baseflow, indicated a strong isotopic fractionation between river icing and stream water (Souchez and Jouzel, 1984). The switch of water head in the deep wells within cluster WW01, from falling to rising in April 2015, may mark the initial melting of river ice and the sharp rise in June 2015 may



mark the beginning of inflow from upstream to the plain aquifer. The latter is consistent in time with the refilling of dry upstream channels with runoff.

## 5.3 Conceptual model of the groundwater exchange and pathways

Based on the geochemical, thermal, isotopic and hydrological results, a conceptual model of the hydrological connectivity in mountain-hill-plain complex was developed (Figure 12). Groundwater in the high mountains mainly occurs as suprapermafrost groundwater within either moraine and scree deposits or surficial fissures in bedrock outcrop areas. In the moraine and fluvio-glacial deposits on the planation surfaces of higher hills (> 3500 m), suprapermafrost, intrapermafrost, and subpermafrost groundwater co-occur. There are three hydrological passages through which meltwater and precipitation are transported from the high mountains to the plain. The first and fastest is the stream channel, which generally originates at the glacier front and is fed by glacial and snow meltwater in its head. Then it moves by overland flow and suprapermafrost groundwater over its course from the mountains to the piedmont sloping plain, and also probably by subpermafrost groundwater at the base of hill slopes. The stream percolates down into the aquifers when flowing through the open plain, and is recharged by groundwater when flowing through the gorge at the north end of the plain. This passage is available only during the warm season and it dries up during the cold season. The second passage is the slope surface and suprapermafrost aquifer, which collect precipitation over a large area, then transport much of it as overland flow and suprapermafrost groundwater into talus fans at the base of mountains or hills, and finally into the aquifer at the top of the plain. Where the moraine and scree deposits in high mountains adjoin the moraine and fluvio-glacial deposits on higher hills, the glacier and snow meltwater may also be transported through this passage after flowing through moraine and scree deposits into the suprapermafrost reservoir at the lower margin of cirques. This passage is also seasonal. The third passage is the subpermafrost aquifer occurring on the planation surface, which conducts meltwater percolating down over the moraines within cirques to the hill slopes, and finally into the aquifer at the top of the plain via a variety of pathways. The water within the second passage is also added to this passage through supra- and subpermafrost connections on the planation surface. This passage is the slowest one, but also the only one that is available during cold season.

The Quaternary aquifer under the piedmont plain is mainly recharged by lateral flow from the south mountains and hills and by the seepage of streams when they flow through the open plain and discharge mainly as baseflow to the stream in the north gorge. The water table dynamics at the top of the plain are characterized by sharp rises and recessions in response to heavy rainfall events but a gradual decline during the cold season. Water table dynamics at the base of the plain are characterized by a stable condition confined to a narrow range. This behavior indicates a rapid transfer of groundwater from the south top to the north base of the plain during the high-flow period and a slow release of stored groundwater during the low-flow period. It suggests that the groundwater under the plain not only maintains stream flow over the cold season, but also contributes significantly to stream flow during the rainy season. We propose two mechanisms involved in the significant seasonal variation of the aquifer in water-conduction capacity. These are surface drainage through the stream channel and



subsurface drainage to an artesian aquifer confined by stream icing and seasonal frost (Figure 12). The first mechanism is similar to "fill and spill" in hillslope hydrology (Spence and Woo, 2003; Tromp-van Meerveld and McDonnell, 2006) and involves the funnel-shaped distribution of unconsolidated permeable deposits on the plain. When moving from the wide plain to the gorge, the cross section of groundwater flow becomes narrow, resulting in reduced transmissivity, increased flow resistance, and an uplifted water table. This ensures that the water table in front of the gorge never drops below the channel bed and thus the downstream channel continuously flows throughout the year. The unchecked surface drainage through the stream channel prevents the water table from rising too high after storms in the rainy season. This mechanism explains the rapid transfer of groundwater from the top to the base of the plain and the stable water table in front of the gorge during the high-flow period. The second mechanism works only during the cold season, when the stream icing and seasonal frost converts the bottom of the gorge into a confined aquifer. The rise of the downstream groundwater head reduces the hydraulic gradient between the wide plain and the gorge, resulting in decreased discharge and reduced transmissivity. In addition, the increased icing constricts the channel cross section while the descending frost reduces the effective thickness of saturated soil, and significantly reduces groundwater discharge into the channel. This mechanism explains the slow release of stored groundwater from the plain and thus the gradual decline of the water table at the top of the plain during the low-flow period.

## 6. Conclusions

Knowledge of groundwater systems in permafrost areas is often meagre (Kane et al., 2013). Groundwater studies in permafrost are challenging given the limited infrastructure and the short field season. These conditions favor samples from baseflow discharge and perennial groundwater springs, combined with the use of geochemical and isotope tracers to elucidate recharge conditions and flow paths. We selected a representative catchment in the headwater region of Heihe, Qinghai-Tibet Plateau as a study site. The study used groundwater head, temperature, geochemical, and isotopic information to determine the roles of groundwater within the permafrost zone for hydrologically connecting waters originating from glaciers in the high mountains to lower elevation rivers.

Previous studies reported that groundwater in the permafrost region occurred only as suprapermafrost groundwater (Cao, 1977). Our field study confirms the co-occurrence of supra-, intra- and subpermafrost groundwater. The suprapermafrost groundwater is mainly recharged by local precipitation and glacier meltwater, and discharged into streams as baseflow, or onto the surface as seeps and springs. An additional discharge route of suprapermafrost groundwater is leakage to the subpermafrost aquifer through sinkholes. The suprapermafrost groundwater generally has a short flow path, leading to a relatively short residence time and weak water-rock interactions. Recharge mostly occurs during warm seasons since the source waters (glacier meltwater and precipitation) are mainly concentrated when the active layer is thawed. Limited recharge occurs in the cold season because the recharge sources and the active layer are frozen. Due to a change in the thawing depth of the active layer and frequent conversion of the recharge-discharge interrelationship, the storage of suprapermafrost groundwater varies significantly throughout the warm seasons. The subpermafrost groundwater on the





planation surface is strongly linked to the surface hydrological processes and it is recharged from suprapermafrost groundwater and glacier and snow meltwater. The chemical and isotopic results indicated that the suprapermafrost groundwater had not flowed through the underlying bedrock and then moved upward to the subpermafrost aquifer after deep circulation along fissures as previous study indicated (Evans et al., 2015). The glacier meltwater recharging the subpermafrost aquifer occurred mainly at localized water bodies. The highly permeable unconsolidated materials in which the subpermafrost groundwater is stored facilitate fast groundwater discharge. The subpermafrost groundwater discharges into streams as baseflow or onto the surface as seeps and springs. The subpermafrost groundwater is also recharged mostly in warm seasons, whereas the recharge is very limited in cold seasons. Intrapermafrost groundwater occurs in a closed hydrochemical talik with strong reducing environments and poor hydraulic connections with the suprapermafrost and subpermafrost groundwater.

The moraine and fluvio-glacial deposits on the planation surfaces of the higher hills are commonly distributed in the head water regions of the Heihe River. These deposits provide another major reservoir for the storage and flow of groundwater in the permafrost region. This is the first report on the occurrence of subpermafrost and intrapermafrost groundwater in the head water regions of the Heihe River.

The groundwater under the piedmont plain within the seasonal frost zone is mainly recharged by lateral flow from the south mountains and hills and the seepage of streams, and is discharged as baseflow into the stream in the north gorge. A rapid transfer of groundwater from the south top to the north base of the plain occurs during the high-flow period whereas stored groundwater is slowly released during the low-flow period. The seasonal variation of the aquifer in water-conduction capacity was interpreted by two mechanisms: (1) surface drainage via the stream channel, similar to "fill and spill" mechanism in hillslope hydrology. The narrowed cross section of groundwater flow from the wide plain to the gorge led to an increase in the water table, preventing the water table upstream from the gorge from dropping below the channel bed and maintaining the continuous flow in the downstream channel throughout the year. This explains the rapid transfer of groundwater from the top to the base of the plain and the stable water table in front of the gorge during the high-flow period; and (2) subsurface drainage to an artesian aquifer confined by stream icing and seasonal frost. When the stream icing and seasonal frost changes the bottom of the gorge into a confined aquifer during the cold season, downstream groundwater head rises and the hydraulic gradient between the wide plain and the narrow gorge is reduced. In addition, increased icing constricts the channel cross section, significantly reducing groundwater discharge into the river channel. The second mechanism proposed here explains the slow release of stored groundwater from the plain and thus the gradual decline of the water table at the top of the plain during the low-flow period. This expanded the existing "fill and spill" mode for catchment and hillslope hydrology.

## Acknowledgements

This research was financially supported by National Natural Science Foundations of China (NSFC-91325101 and 91125009)



and the Grant for Innovative Research Groups of the National Natural Science Foundation of China (41521001).

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



**Figure captions:**

**Figure 1.** (a) Location of the head water regions of Heihe River Basin and the distribution of Quaternary unconsolidated deposits within the upper Heihe River Basin; (b) Hulugou Catchment study area showing monitoring and sampling sites; (c) geological cross section.

**Figure 2.** (a) Precipitation recorded at an elevation of 3649 masl, and (b) air temperature within the Hulugou Catchment from April 2014 to October 2015.

**Figure 3.** The well log of the sediments for cluster well WW01, WW02 and WW03 within the seasonal frost zone, and WW04 within the permafrost zone.

**Figure 4.** Water table depth in the cluster wells WW03 at the up-gradient recharge zone and WW01 at the down-gradient discharge zone within the seasonal frost zone, and in the cluster well WW04 within permafrost zone.

**Figure 5**. Temperature envelopes in the sediments at locations of WW01, WW02 and WW03 within the seasonal frost zone, and WW04 within the permafrost zone.

**Figure 6.** The piper diagram for groundwaters, river waters, rain and snow-melting waters in the study area. H refers to summer high flows; L refers to winter low flows.

**Figure 7.** Groundwater chemistry at different depths of various cluster wells within seasonal frost zone (a), and chemistry of groundwater, river water and thermokarst water within permafrost zone (b). WW01, WW03 and WW04 are symbols for cluster wells. The number in brackets means the specific screen depth of cluster wells. Note the log scale on the y-axis.

**Figure 8.** The $\delta^{18}$O and $\delta$D relationship in different water types and in permafrost sediments from September 2014 to August 2015.

**Figure 9.** The change of $\delta^{18}$O and $\delta$D compositions along depth of iced sediments below ground surface within permafrost zone.

**Figure 10.** Heihe River flux (a), and the $\delta^{18}$O and $\delta$D values change with time in river waters (b), well waters (c), and spring waters (d) from June 2014 to May 2016.

**Figure 11.** Photos of sinkholes widely developed within permafrsot zone (a), and river icing (left) and spring icing (right) at the study site in winter (b).

**Figure 12.** Conceptual model of groundwater flow system at the study site.



**Table 1.** Mean values, standard deviations (± SD) and number of samples used to determine chemical concentrations in different types of waters from July to September, and from January to April in 2014-2015. H refers to summer high flows; L refers to winter low flows. n.s. means that no samples were collected.

| Water type | Sample No. | Sample number H | L | Ca²⁺ H | Ca²⁺ L | Mg²⁺ H | Mg²⁺ L | Na⁺ H | Na⁺ L | K⁺ H | K⁺ L | Sr H | Sr L | Si H | Si L |
|---|---|---|---|---|---|---|---|---|---|---|---|---|---|---|---|
| East tributary, periglacial and permafrost zone | RW27 | 18 | 2 | 28.5±3.9 | 11.3±2.7 | 15.6±2.3 | 5.3±1.3 | 1.6±0.4 | 0.9±0.1 | 0.5±0.1 | 0.3±0.1 | 0.2±0.0 | 0.1±0.0 | 0.9±0.1 | 0.2±0.0 |
| | RW28 | 15 | 2 | 27.9±2.7 | 16.5±9.1 | 15.2±1.5 | 8.4±4.2 | 1.6±0.5 | 1.3±0.5 | 0.5±0.2 | 0.4±0.1 | 0.2±0.0 | 0.1±0.0 | 0.9±0.1 | 0.3±0.3 |
| | RW03 | 15 | 0 | 27.3±2.6 | n.s. | 14.5±1.7 | n.s. | 1.5±0.4 | n.s. | 0.4±0.1 | n.s. | 0.2±0.0 | n.s. | 0.8±0.1 | n.s. |
| East tributary, seasonal frost zone | RW29 | 17 | 2 | 34.0±7.3 | 19.3±13.3 | 20.7±6 | 14.2±10.0 | 2.5±1.2 | 1.8±1.4 | 0.6±0.1 | 0.4±0.2 | 0.2±0.1 | 0.1±0.1 | 1.2±0.3 | 0.4±0.3 |
| | RW30 | 15 | 0 | 33.6±9.6 | n.s. | 20.0±6.9 | n.s. | 2.4±1.3 | n.s. | 0.5±0.1 | n.s. | 0.2±0.1 | n.s. | 1.1±0.2 | n.s. |
| West tributary, seasonal frost zone | RW24 | 19 | 3 | 26.1±1.8 | 32.5±14.4 | 12.6±1.0 | 16.7±5.9 | 1.8±0.3 | 2.1±0.6 | 0.6±0.3 | 0.7±0.4 | 0.1±0.0 | 0.2±0.1 | 1.1±0.1 | 1.4±1.1 |
| | RW25 | 19 | 0 | 31.2±2.8 | n.s. | 15.1±1.8 | n.s. | 2.2±0.4 | n.s. | 0.6±0.1 | n.s. | 0.1±0.0 | n.s. | 1.2±0.1 | n.s. |
| | RW26 | 19 | 0 | 32.3±3.1 | n.s. | 15.5±1.8 | n.s. | 2.5±0.5 | n.s. | 0.7±0.1 | n.s. | 0.1± 0.0 | n.s. | 1.2±0.1 | n.s. |
| Catchment outlet, seasonal frost zone | RW08 | 20 | 3 | 46.6±8.9 | 60.2±37.7 | 28.5±6.3 | 39.3±23.2 | 4.9±1.7 | 9.7±6.9 | 0.8±0.2 | 1.2±0.8 | 0.3±0.1 | 0.5±0.4 | 1.6±0.2 | 1.6±1.2 |
| | RW10 | 20 | 4 | 50.9±6.5 | 76.7±11.2 | 31.3±4.6 | 36.6±4.3 | 7.3±1.6 | 26.2±12.2 | 0.9±0.2 | 1.6±0.3 | 0.4±0.1 | 0.8±0.1 | 1.8±0.2 | 2.8±1.3 |
| Spring | QW02 | 19 | 4 | 59.3±4.1 | 51.4±16.1 | 36.7±1.7 | 31.5±7.8 | 14.1±0.8 | 12.4±3.9 | 1.4±0.2 | 1.1±0.6 | 0.5±0.0 | 0.5±0.2 | 2.3±0.2 | 1.7±1.3 |
| | QW03 | 20 | 4 | 62.7±5.7 | 48.6±14.1 | 37.5±2.6 | 30.3±8.0 | 16.6±0.4 | 13.4±6.2 | 1.4±0.3 | 1.0±0.5 | 0.6±0.0 | 0.5±0.2 | 2.4±0.3 | 1.9±1.0 |
| | QW04 | 20 | 4 | 64.5±5.1 | 48.5±15.3 | 39.0±2.3 | 30.4±8.6 | 18±0.3 | 13.5±5.5 | 1.5±0.3 | 1.1±0.6 | 0.6±0.0 | 0.5±0.2 | 2.4±0.2 | 1.8±1.2 |
| | QW05 | 20 | 4 | 65.5±7.3 | 52.9±10.0 | 39.7±3.2 | 34.0±5.7 | 19.1±0.5 | 16.3±4.2 | 1.5±0.3 | 1.2±0.4 | 0.7± 0.0 | 0.6±0.1 | 2.6±0.8 | 2.0±1.1 |
| | QW08 | 20 | 4 | 58.9±5.5 | 44.6±15.2 | 36.2±3.0 | 28.2±8.2 | 13±1.3 | 8.0±2.6 | 1.1±0.1 | 0.9±0.5 | 0.6±0.0 | 0.5±0.2 | 2.3±0.2 | 1.7±1.2 |
| Well water at permafrost zone | WW04 (24.3m) | 1 | 0 | 47.4 | n.s. | 22.9 | n.s. | 23.3 | n.s. | 6.4 | n.s. | 0.3 | n.s. | 1.3 | n.s. |
| | WW04 (12m) | 0 | 2 | n.s. | 204.6±1.4 | n.s. | 95.9±1.3 | n.s. | 221.0±10.4 | n.s. | 9.7±1.8 | n.s. | 2.7±0.1 | n.s. | 9.1± 0.2 |
| | WW04 (1.5m) | 17 | 0 | 72.4±5.7 | n.s. | 15.4±1.4 | n.s. | 8.6±2.8 | n.s. | 4.3±1.3 | n.s. | 0.3±0.0 | n.s. | 3.9±0.3 | n.s. |
| Well water at seasonal frost, upgradient | WW03 (30m) | 19 | 4 | 55.1±11 | 33.3±13.2 | 36.7±6.3 | 25.2±9.9 | 17.6±15.1 | 21.1±10.6 | 1.6±0.9 | 1.7±1.0 | 0.4±0.1 | 0.3±0.1 | 1.8±0.1 | 1.2±1.2 |
| | WW03 (20m) | 19 | 3 | 60.1±6.9 | 57.0±9.6 | 38.8±3.8 | 35.6±7.2 | 7.3±2.7 | 6.7±1.3 | 1.2±0.4 | 1.4±0.4 | 0.4±0.1 | 0.4±0.1 | 1.9±0.2 | 1.9±1.0 |
| | WW03(10m) | 6 | 0 | 46.5±11.8 | n.s. | 35.7±13.1 | n.s. | 9.4±6.8 | n.s. | 4.7±5 | n.s. | 0.3±0.1 | n.s. | 1.9±0.5 | n.s. |
| Well water at seasonal frost, downgradient | WW01 (25 m) | 19 | 4 | 65.3±20.1 | 31.4±4.3 | 43.2±10.9 | 20.7±9.9 | 24.5±42.5 | 14.5±7.7 | 1.7±1.2 | 1.3±0.8 | 0.5±0.1 | 0.2±0.1 | 2.2±0.3 | 1.1±1.1 |
| | WW01 (15 m) | 19 | 4 | 67.1±15.7 | 70.8±20.1 | 43.8±9.6 | 41.8±11.3 | 13.2±18.1 | 13.1±7.8 | 1.4±0.7 | 1.5±0.7 | 0.5±0.1 | 0.5±0.1 | 2.2±0.3 | 1.6±0.9 |
| | WW01 (10 m) | 19 | 2 | 64.9±17.4 | 80.8±3.6 | 42.2±11.8 | 34.4±4.8 | 8.7±2.2 | 6.6±1.3 | 1.3±0.6 | 0.7±0.2 | 0.4±0.1 | 0.4±0.0 | 2±0.4 | 0.4±0.2 |
| | WW01 (5 m) | 12 | 0 | 76.6±8.4 | n.s. | 48.9±4.8 | n.s. | 9.0±1.0 | n.s. | 1±0.1 | n.s. | 0.5±0.1 | n.s. | 2.1±0.1 | n.s. |



| Water type | | Sample number | | $SO_4^{2-}$ | | $NO_3^-$ | | $Cl^-$ | | $HCO_3^-$ | | TDS | |
|---|---|---|---|---|---|---|---|---|---|---|---|---|---|
| Location | Sample No. | H | L | H | L | H | L | H | L | H | L | H | L |
| East tributary, periglacial and permafrost zone | RW27 | 18 | 2 | 29.5±11 | 32.9±13.6 | 2.2±1.1 | 1.4±0.6 | 3.3±0.1 | 7.3±0.9 | 105.6±13.2 | 104.0±4.1 | 134±16.7 | 111.5±11.8 |
| | RW28 | 15 | 2 | 31.7±8.3 | 30.2±11.0 | 2.5±0.9 | 1.4±0.2 | 3.1±0.3 | 8.1±2.5 | 102.1±12.9 | 107.3±5.5 | 133.6±14.2 | 120.0±30.1 |
| | RW03 | 15 | 0 | 31.2±8.9 | n.s. | 2.6±0.9 | n.s. | 3.3±0.2 | n.s. | 101.2±12.2 | n.s. | 131.5±14.4 | n.s. |
| East tributary, seasonal frost zone | RW29 | 17 | 2 | 45.3±16.6 | 57.6±14.2 | 2.9±1 | 1.7±0.4 | 3.4±0.4 | 6.6±0.0 | 131.4±29.8 | 133.6±9.9 | 175.1±44.9 | 168.6±44.4 |
| | RW30 | 15 | 0 | 46.8±27.1 | n.s. | 3±1.2 | n.s. | 3.3±0.4 | n.s. | 125.1±33.2 | n.s. | 172.1±61.0 | n.s. |
| West tributary, seasonal frost zone | RW24 | 19 | 3 | 28.7±8.4 | 73.3±14.8 | 2.7±0.6 | 1.9±0.4 | 3.5±0.2 | 3.9±0.2 | 96±6.9 | 122.7±5.8 | 124.2±11.5 | 192.5±34.0 |
| | RW25 | 19 | 0 | 41.4±10.1 | n.s. | 3.1±0.7 | n.s. | 3.7±0.2 | n.s. | 108.5±11.5 | n.s. | 151.6±17.6 | n.s. |
| | RW26 | 19 | 0 | 44.7±10.6 | n.s. | 3.2±0.7 | n.s. | 3.6±0.3 | n.s. | 108±9.5 | n.s. | 156.7±18 | n.s. |
| Catchment outlet, seasonal frost zone | RW08 | 20 | 3 | 92±30 | 207.9±107.0 | 3.9±1.2 | 3.1±1.1 | 3.7±0.4 | 5.2±0.6 | 166.3±30 | 240.0±85.6 | 263.6±61.2 | 446.8±218.8 |
| | RW10 | 20 | 4 | 106.4±23.8 | 246.1±71.0 | 4±1.1 | 2.7±0.7 | 4±0.4 | 6.9±2.3 | 178.8±25.7 | 238.0±4.5 | 294.4±47.5 | 516.1±89.7 |
| Spring | QW02 | 19 | 4 | 126.1±27.7 | 162.1±31.1 | 4.2±1.4 | 2.8±0.4 | 4.9±0.5 | 5.5±0.4 | 222.3±12.8 | 230.7±2.6 | 358±31.4 | 382.4±22.7 |
| | QW03 | 20 | 4 | 137.3±43.6 | 152.7±26.8 | 4.1±1.6 | 2.6±0.7 | 5±0.7 | 5.3±0.7 | 233.1±13.9 | 238.8±3.4 | 381.2±50.9 | 373.4±56.8 |
| | QW04 | 20 | 4 | 150.1±34.1 | 156.0±20.0 | 4.2±1.4 | 2.8±0.4 | 5.6±0.6 | 5.4±0.4 | 237.7±10.7 | 240.3±4.7 | 401.9±38 | 378.2±48.6 |
| | QW05 | 20 | 4 | 150.9±34.2 | 147.7±39.1 | 4.2±1.5 | 5.3±6.6 | 5.5±0.6 | 5.3±0.8 | 243.4±11.2 | 248.7±5.5 | 408.4±37.2 | 387.3±53.8 |
| | QW08 | 20 | 4 | 95.4±33.2 | 107.5±37.1 | 4.3±1.8 | 2.4±1.0 | 4.4±0.5 | 4.6±0.7 | 254.8±14.1 | 246.1±6.0 | 340.8±40.3 | 319.5±61.1 |
| Well water at permafrost zone | WW04 (24.3m) | 1 | 0 | 64.7 | n.s. | 1.5 | n.s. | 17.6 | n.s. | 237.5 | n.s. | 302.9 | n.s. |
| | WW04 (12m) | 0 | 2 | n.s. | 4.1±0.0 | n.s. | 0.2±0.2 | n.s. | 106.4±10.4 | n.s. | 833.6±30.2 | n.s. | 1059.3±40.6 |
| | WW04 (1.5m) | 17 | 0 | 10.2±5.5 | n.s. | 0.3±1.1 | n.s. | 6.1±1.1 | n.s. | 294.3±25.7 | n.s. | 264.6±18.2 | n.s. |
| Well water at seasonal frost, upgradient | WW03 (30m) | 19 | 4 | 115±16.8 | 74.9±24.6 | 3.8±1.3 | 0.1±0.1 | 4.5±0.9 | 4.9±0.3 | 243.8±24.9 | 282.4±14.7 | 356.4±29.6 | 302.6±52.1 |
| | WW03 (20m) | 19 | 3 | 116.2±31.7 | 148.8±26.5 | 4±1.5 | 2.4±0.4 | 4±0.5 | 5.1±0.4 | 236.1±31.4 | 250.2±9.0 | 349.7±41.3 | 382.3±42.1 |
| | WW03(10m) | 6 | 0 | 122.4±64.2 | n.s. | 3.6±0.7 | n.s. | 5±1.1 | n.s. | 297.8±136.3 | n.s. | 409.5±196.4 | n.s. |
| Well water at seasonal frost, downgradient | WW01 (25 m) | 19 | 4 | 158.9±51.6 | 92.4±22.0 | 4.9±2.4 | 0.7±0.7 | 5.2±2.7 | 5.1±0.4 | 270.6±43 | 212.1±32.5 | 439.2±95.9 | 272.5±61.0 |
| | WW01 (15 m) | 19 | 4 | 162.4±33.6 | 212.3±70.9 | 5.5±2 | 2.8±0.9 | 4.8±1.1 | 5.0±0.2 | 255±42.2 | 287.6±58.4 | 425.9±78.2 | 491.3±126.7 |
| | WW01 (10 m) | 19 | 2 | 150.5±52.6 | 267.7±26.6 | 5.3±2.1 | 2.6±0.2 | 4.7±0.7 | 5.1±0.4 | 243.2±54.6 | 332.2±2.7 | 399.4±109.3 | 564.1±35.9 |
| | WW01 (5 m) | 12 | 0 | 172.6±33.2 | n.s. | 6.9±1.2 | n.s. | 4.5±0.3 | n.s. | 269.4±23.9 | n.s. | 454.3±58 | n.s. |





**Table 2.** $^3$H, $^{13}$C and $^{14}$C isotopic composition of groundwater samples and $^{14}$C ages corrected using $^{13}$C mixing model.

| Sample No. | $^3$H | $\delta^{13}$C (‰) | | pmC (%) | | $^{14}$C Age (a BP) | | Corrected age with $^{13}$C mixing model | |
|---|---|---|---|---|---|---|---|---|---|
| | | $\delta^{13}$C | Error (1σ) | pmC | Error (1σ) | $^{14}$C Age | Error (1σ) | q | corrected age(yr) |
| No.4 (24.3 m well) | | -16.77 | 0.51 | 76.43 | 0.32 | 2159 | 34 | 0.9 | 1637 |
| No.4 (1.5 m well) | 15.11 | -13.6 | 0.57 | 96.34 | 0.31 | 299 | 26 | 0.76 | -2009(modern) |
| No.3 (30 m well) | 19.38 | -8.79 | 0.57 | 51.77 | 0.22 | 5288 | 33 | 0.49 | -483 (modern) |
| No.3 (20 m well) | 16.22 | n.d. | n.d. | n.d. | n.d. | n.c. | n.c. | n.c. | n.c. |
| QWIP01 | 20.69 | -8.31 | 0.61 | 35.51 | 0.17 | 8317 | 39 | 0.46 | 2170 |
| QWIP02 | 17.33 | -8.05 | 0.55 | 49.6 | 0.2 | 5632 | 32 | 0.45 | -856 (modern) |
| No.1 (25 m well) | 16.95 | -5.92 | 0.53 | 44.38 | 0.18 | 6525 | 33 | 0.33 | -2477(modern) |
| No.1 (15 m well) | 24.18 | n.d. | n.d. | n.d. | n.d. | n.c. | n.c. | n.c. | n.c. |
| No.1 (10 m well) | 16.20 | n.d. | n.d. | n.d. | n.d. | n.c. | n.c. | n.c. | n.c. |
| QW02 | 27.83 | n.d. | n.d. | n.d. | n.d. | n.c. | n.c. | n.c. | n.c. |
| QW03 | 13.84 | n.d. | n.d. | n.d. | n.d. | n.c. | n.c. | n.c. | n.c. |
| QW05 | 43.59 | n.d. | n.d. | n.d. | n.d. | n.c. | n.c. | n.c. | n.c. |
| QW04 | 13.61 | -5.09 | 0.7 | 43.05 | 0.19 | 6770 | 34 | 0.28 | -3475(modern) |
| QW08 | 18.58 | n.d. | n.d. | n.d. | n.d. | n.c. | n.c. | n.c. | n.c. |

n.d. means not determined and n.c. means not calculated.





**Figure 1.** (a) Location of the head water regions of Heihe River Basin and the distribution of Quaternary unconsolidated deposits within the upper Heihe River Basin; (b) Hulugou Catchment study area showing monitoring and sampling sites; (c) geological cross section.





**Figure 2.** (a) Precipitation recorded at an elevation of 3649 masl, and (b) air temperature within the Hulugou Catchment from April 2014 to October 2015.





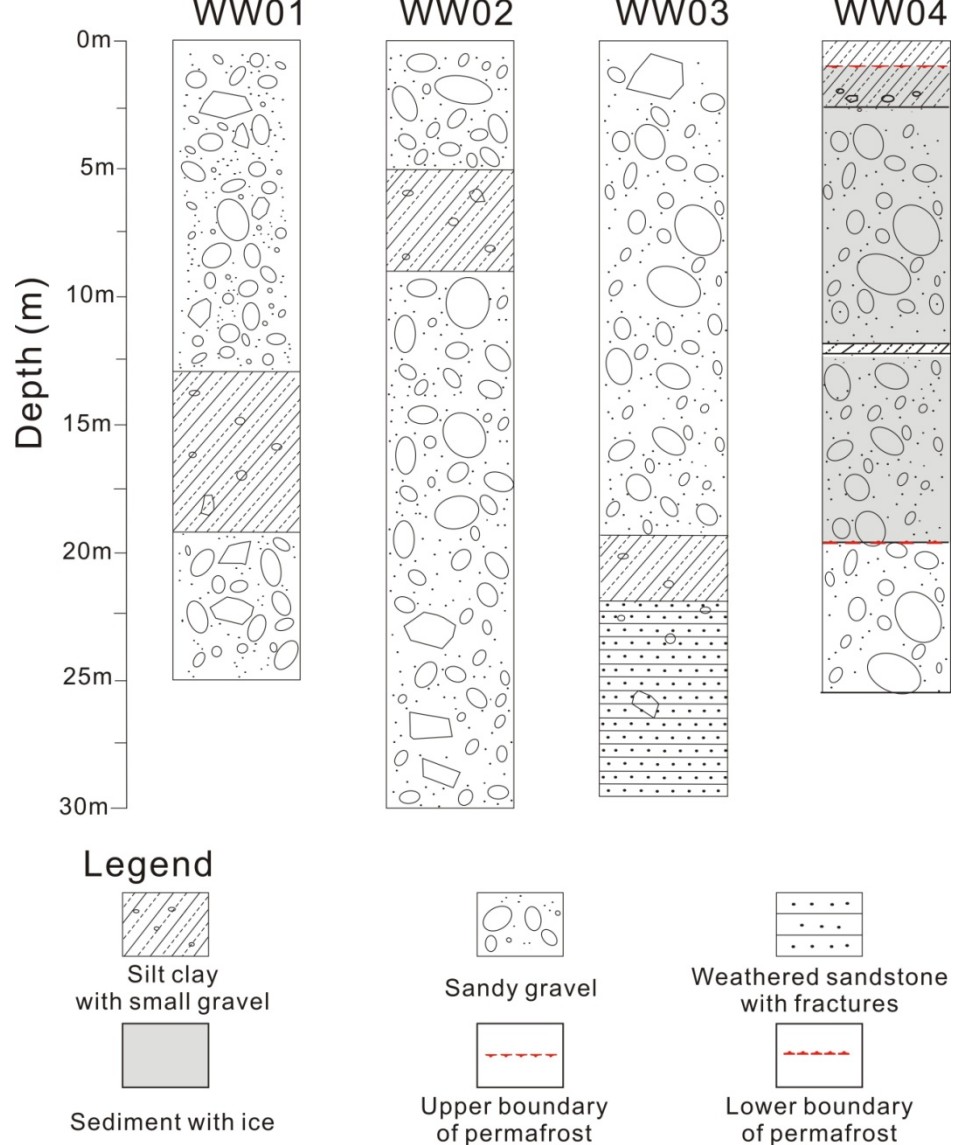

**Figure 3.** The well log of the sediments for cluster well WW01, WW02 and WW03 within the seasonal frost zone, and WW04 within the permafrost zone.





**Figure 4.** Water table depth in the cluster wells WW03 at the up-gradient recharge zone and WW01 at the down-gradient discharge zone within the seasonal frost zone, and in the cluster well WW04 within permafrost zone.





**Figure 5**. Temperature envelopes in the sediments at locations of WW01, WW02 and WW03 within the seasonal frost zone, and WW04 within the permafrost zone.





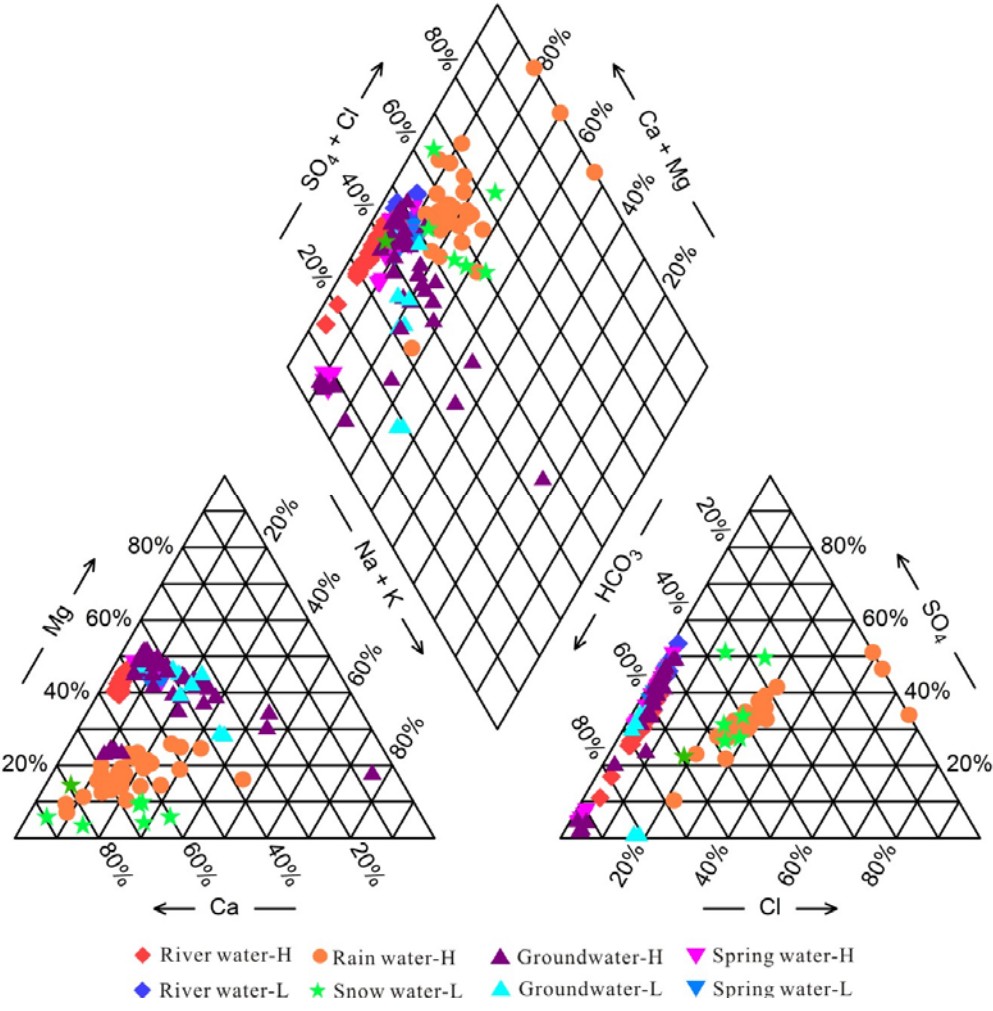

**Figure 6.** The piper diagram for groundwaters, river waters, rain and snow-melting waters in the study area. H refers to summer high flows; L refers to winter low flows.





**Figure 7.** Groundwater chemistry at different depths of various cluster wells within seasonal frost zone (a), and chemistry of groundwater, river water and thermokarst water within permafrost zone (b). WW01, WW03 and WW04 are symbols for cluster wells. The number in brackets means the specific screen depth of cluster wells. Note the log scale on the y-axis.





**Figure 8.** The $\delta^{18}$O and $\delta$D relationship in different water types and in permafrost sediments from September 2014 to August 2015.




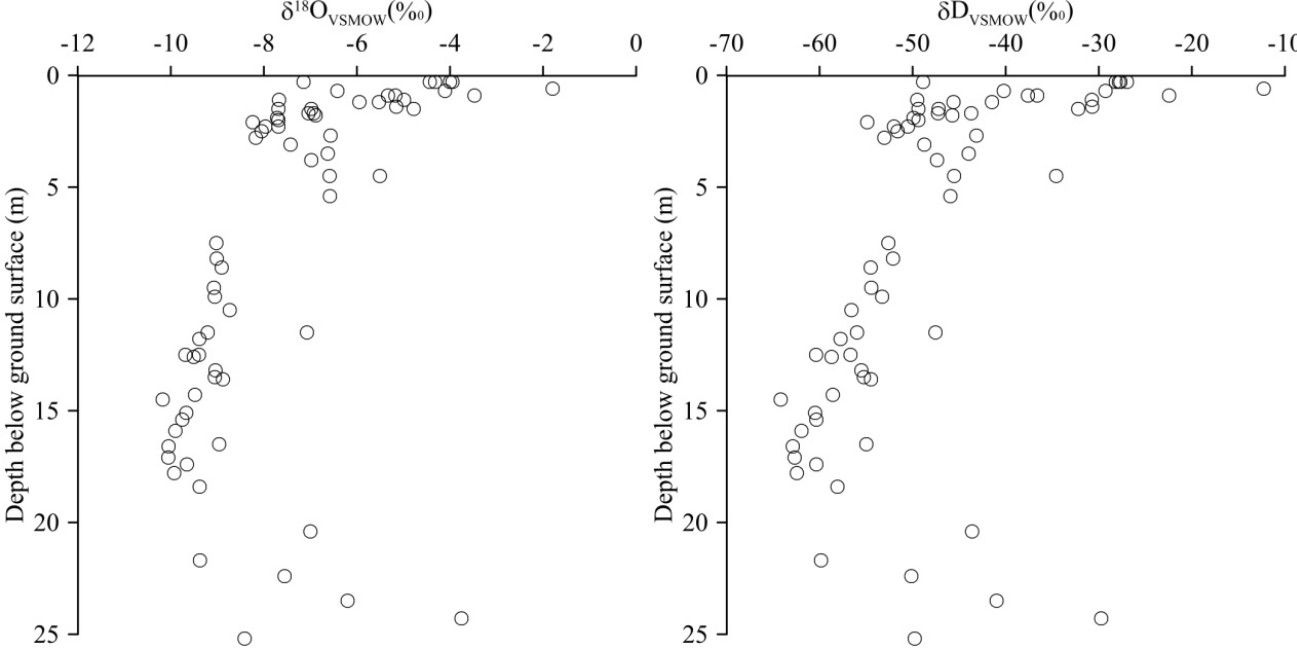

**Figure 9.** The change of $\delta^{18}$O and $\delta$D compositions along depth of iced sediments below ground surface within permafrost zone.




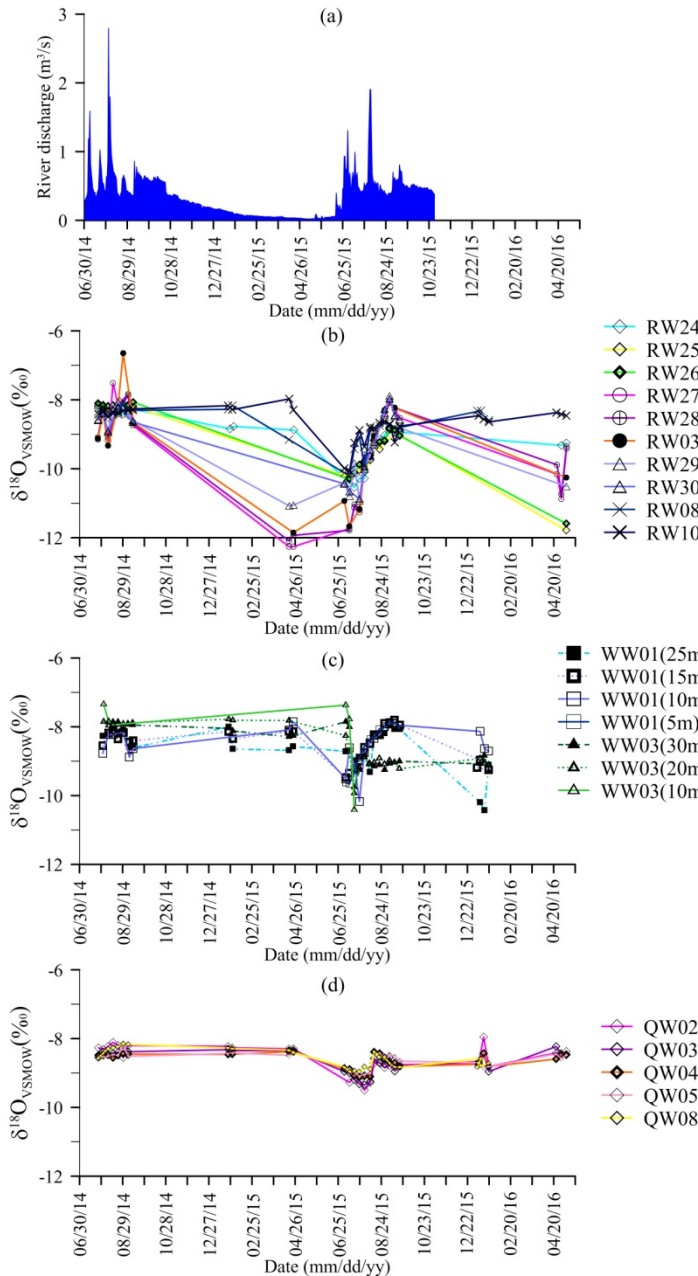

**Figure 10.** Heihe River flux (a), and the $\delta^{18}$O and $\delta$D values change with time in river waters (b), well waters (c), and spring waters (d) from June 2014 to May 2016.




**Figure 11.** Photos of sinkholes widely developed within permafrsot zone (a), and river icing (left) and spring icing (right) at the study site in winter (b).


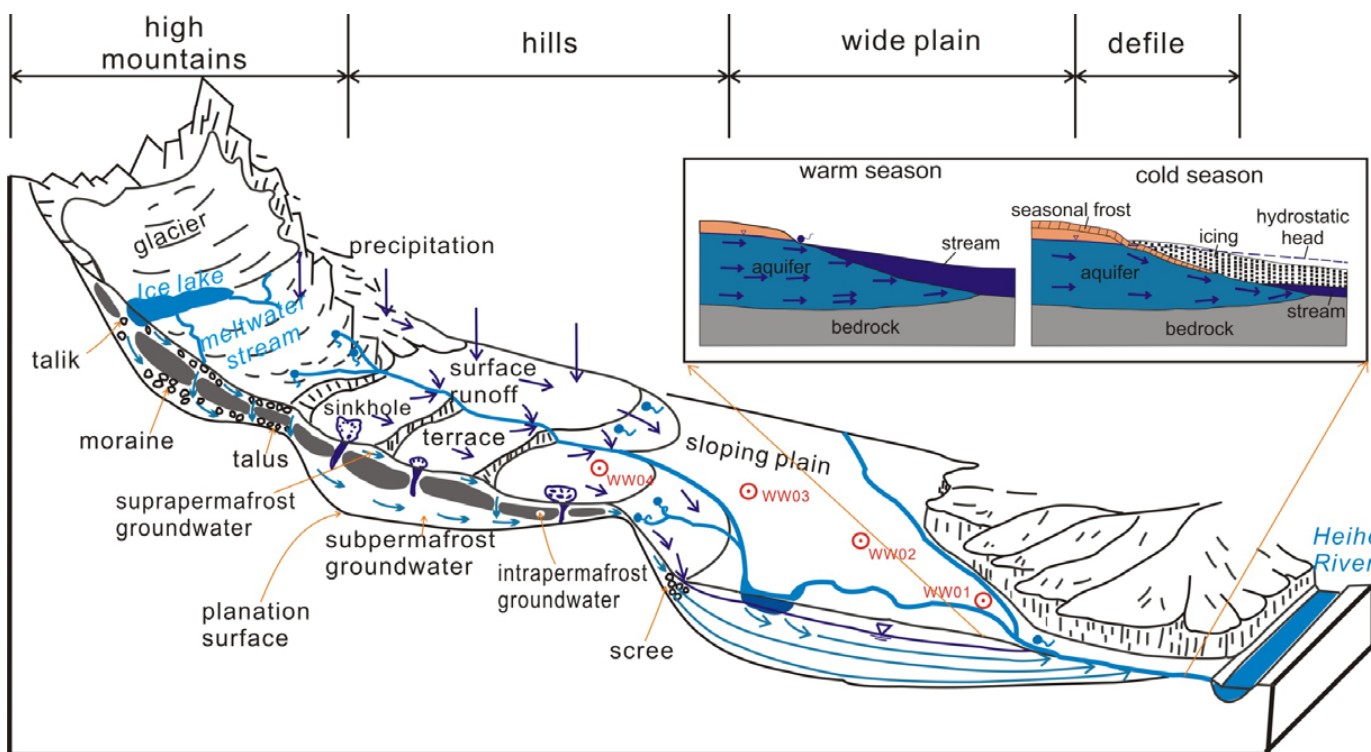

**Figure 12.** Conceptual model of groundwater flow system at the study site.