# Peer review of "Hydrological connectivity from glaciers to rivers in the Qinghai-Tibet Plateau: roles of suprapermafrost and subpermafrost groundwater"

_Hydrology and Earth System Sciences, 2017_

## Referee Comment (RC1) · Anonymous Referee #1 · 4 Mar 2017

General Comments This study addresses an important issue of hydrological connectivity between glaciers in high mountains and river in the low plain within the alpine headwater catchment with big elevation difference and complex hydrogeological settings. The hydraulic head, temperature, and chemical and isotopic composition of groundwater, streamflow, precipitation and glacier meltwater were monitored along altitude gradient. The work has produced a remarkably rich data set that is clearly presented by the authors. The authors interpret the data to indicate that supra- and subpermafrost aquifers, as well as stream channels and slope surfaces, play an important role in transporting glacier, snow-meltwater and precipitation from the high mountains to the plain and then to the mainstem. The authors also suggest that a decline in hydrological connectivity between the piedmont plain aquifer and the downstream channel in cold seasons may be the mechanism maintaining streamflow (baseflow) in winter. It is worth pointing out that the authors present a logical and clearly illustrated conceptual model of hydrological connectivity in the alpine catchment by combining the above results. Given the wide distribution of this kind of headwater "mountain-plain-river" catchments in the Qinghai-Tibet Plateau and other cold regions, this conceptual model may contribute fundamentally to permafrost hydrology and can be more broadly utilized. The authors tentatively suggest that river icing and riverbank soil freezing may form a confining layer to reduce groundwater discharge from the plain to the stream, i.e., reduce the hydrological connectivity between the two pools. This is a very interesting hypothesis that can expand the existing mode for interpreting the slow release of stored groundwater during cold seasons, and it may be testable using field hydrometric measurement and numerical simulation. Overall the manuscript is well written and quite clear. I also have a few minor comments that I hope the authors to address before publication as listed in below.

Specific Comments P2, L5: 'surface-water' should be 'surface water'.

P2, L24 and L25: Two ';'s after 'hydrogeological' should be type errors.

P3, L6: 'Heihe Basin' should be 'Heihe River Basin'.

P3, L22: 'Qinghai-Tibet plateau' should be 'Qinghai-Tibet Plateau'.

P4, L15: Is 'the October to May cold season' a type error? 'ice covered' should be 'ice-covered'.

P6, L14: Citation is missing for the Gran titration method.

P7, L27: What value does the $\delta 13C_{ech}$ take?

P8, L22-23: This sentence is hard to understand. Please rewrite it.

P10, L15: Two 'respectively' should be removed.

P11, L9-17: These results contrast with the statements in Abstract section.

P13, L9: 'in water' should be 'in water table'.

P13, L29: I don't think that the dry sediment layer at depths between 12 m and 12.5 m is related to the subpermafrost groundwater.

P13, L29-31: Citation is missing for this statement.

P14, L24-25: Citation is missing for this statement.

---

## Referee Comment (RC2) · Anonymous Referee #2 · 30 Mar 2017

The authors studied the role of permafrost in controlling groundwater flow and the hydrological connections between glaciers in high mountain and river in the low plain with hydraulic head, temperature, geochemical, and isotopic data. The paper is generally well written, and should be of very interest to the research community. My detailed comments are as below: 1. Legend of Fig.12 should be explained clearly, such as the status of runoff (groundwater, surface water) should be depicted. 2. The resolution and framework of Fig. 10 should be improved. 3. The conclusions need to be improved, the author should tell the most important conclusion by the simple statement at this part. 4. Page 11, the value of $\delta 2H$ and $\delta 18O$ indicate that suprapermafrost groundwater had experienced strong evaporation, but the hydrogeochemistyr also

suggest the suprapermafrost groundwater has rapid flow. It should be explained more clearly. 5. The English of the whole manuscript need to be improved.

Please also note the supplement to this comment:
http://www.hydrol-earth-syst-sci-discuss.net/hess-2017-7/hess-2017-7-RC2-supplement.pdf

―――――――――――――――――――

---

## Author Comment (AC2) · 12 Apr 2017

**General comment:**
The authors studied the role of permafrost in controlling groundwater flow and the hydrological connections between glaciers in high mountain and river in the low plain with hydraulic head, temperature, geochemical, and isotopic data. The paper is generally well written, and should be of very interest to the research community.

**Response:** We thank the reviewer for carefully evaluating our manuscript and for constructive and helpful comments and suggestions. Our responses to the specific comments are provided in a point-by-point reply given below.

**Comment 1:** Legend of Fig.12 should be explained clearly, such as the status of runoff (groundwater, surface water) should be depicted.

**Response:** We have revised Fig. 12 (as shown below) to make it clearer. Since there are many symbols in the figure, it is somewhat inconvenient and unclear if explain them in figure caption. Thus, we added a legend in Fig.12 to explain the meaning of different symbols used in the figure.

[Figure]

**Comment 2:** The resolution and framework of Fig. 10 should be improved.

**Response:** The resolution of Fig.10 is already 600 dpi. However, for comparison, we used the same Y-axis scale for three sub-plots (b), (c) and (d). This is the main reason why five lines in sub-plot (c) is too close to distinguish clearly. The sub-plot (c) would have been clearer and more aesthetic if a smaller scale was used for Y-axis. However, given that this figure is designed to show the difference in spatiotemporal variations of $\delta^{18}O$ between three water pools (i.e., shallow groundwater, spring and stream), further providing insights on their hydrology such as response patterns and water sources, this framework can yield more valuable information compared to that with varying scales. For example, as mentioned in the manuscript (P11, L2-5), spring

waters showed the smallest variation in $\delta^{18}O$ among three water pools, indicating a weaker linkage with surface water, and probably a larger recharge area or/and a longer residence time (in well-mixed).

**Comment 3**: The conclusions need to be improved, the author should tell the most important conclusion by the simple statement at this part.

**Response:** We have revised the section Conclusions to focus on the most important things. The revised conclusions is as following:

*Groundwater studies in permafrost area are challenging given the limited infrastructure and the short field season. These conditions favor samples from baseflow discharge and perennial groundwater springs, combined with the use of geochemical and isotope tracers to elucidate recharge conditions and flow paths. We selected a representative catchment in the headwater region of Heihe River, Qinghai-Tibet Plateau as a study site. The study used groundwater head, temperature, geochemical, and isotopic information to determine the roles of groundwater within the permafrost zone for hydrologically connecting waters originating from glaciers in the high mountains to lower elevation rivers.*

*Our field measurements confirm the co-occurrence of supra-, intra- and subpermafrost groundwater in the headwater regions of Heihe River. To our knowledge, this is the first report on the occurrence of subpermafrost and intrapermafrost groundwater in the region. The moraine and fluvio-glacial deposits on the planation surfaces of higher hills provide a major reservoir for the storage and flow of subpermafrost and intrapermafrost groundwater. The subpermafrost groundwater on the planation surface is interconnected to the surface hydrological processes and recharged by suprapermafrost groundwater and glacier and snow meltwater.*

*Glacier and snow-meltwater are transported from the high mountains to the plain through stream channels, slope surfaces, and supra- and subpermafrost aquifers. The groundwater under the piedmont plain within seasonal frost zone is mainly recharged by lateral flow from the south mountains and hills and by the seepage of streams, and is discharged as baseflow into the stream in the north gorge. A rapid transfer of groundwater from the south top to the north base of the plain occurs during the high-flow period, while the stored groundwater is slowly released during the low-flow period. This seasonal variation of the aquifer in water-conduction capacity was interpreted by two mechanisms: (1) surface drainage via the stream channel, analogous to "fill and spill" mechanism in hillslope hydrology. The narrowing of aquifer from the wide plain to the gorge led to a relatively high water table near the gorge, preventing it from dropping below the channel bed and maintaining a perennial flow in the downstream. This also explains the rapid transfer of groundwater from the top to the base of the plain and the stable water table in front of the gorge during the high-flow period; and (2) subsurface drainage to an ephemeral artesian aquifer confined by stream icing and seasonal frost. When the stream icing and seasonal frost changes the bottom of the gorge into a confined aquifer during the*

*cold season, downstream groundwater head rises and the hydraulic gradient between the wide plain and the narrow gorge is reduced. In addition, increased icing constricts the channel cross section, significantly reducing groundwater discharge into the river channel. The second mechanism proposed here explains the slow release of stored groundwater from the plain during the low-flow period. This expanded the existing "fill and spill" mode for catchment and hillslope hydrology."*

**Comment 4:** Page 11, the value of $\delta^2H$ and $\delta^{18}O$ indicate that suprapermafrost groundwater had experienced strong evaporation, but the hydrogeochemistry also suggest the suprapermafrost groundwater has rapid flow. It should be explained more clearly.

**Response:** We have revised this part (P11, L22-27) as suggested to express it more clearly. The revised statement is as following:

*The low TDS, $Cl^-$ and $Na^-$ concentrations and the $HCO_3$-Ca water type suggest that suprapermafrost groundwater had experienced insufficient water-mineral interaction, probably caused by a relatively short residence time. This is supported by the highest $^{14}C$ activity in the suprapermafrost groundwater among all samples, which was 96.34 pmC and close to the atmospheric value (Clark and Fritz, 1997), and a 15.11 TU $^3H$ concentration, which is an indicator of modern water (Zhai et al., 2013). Though occurring on a relatively flat planation surface, the suprapermafrost groundwater is actually easy to drain because the planation surface adjoins the lower slopes in three directions. Add to that the fact that suprapermafrost aquifer is fairly thin whereas rich in organic matter with high permeability, one can understand why it may have a high renewal rate. However, the enriched $^2H$ and $^{18}O$ isotopes, along with samples' position relative to the LMWL in the $\delta^2H$ vs. $\delta^{18}O$ plot, indicate that suprapermafrost groundwater had also experienced a certain degree of evaporation (Figure 8). These two conclusions are not in conflict when considering the high local evaporation (376−650 mm/yr) and shallow suprapermafrost groundwater depth (0−1.5 m below ground). The high groundwater table may also result in very shallow flowpaths for the majority of the water and few possibilities for chemical reactions between the discharging water and the deep mineral soil (Frey et al., 2007; Stotler et al., 2009;Vonk et al., 2015).*

**Comment 5:** The English of the whole manuscript need to be improved.

**Response:** We have tried our best to edit the English and we have also asked the professional English editing service to polish the English writing.

---

## Author Response (AR1)

**Response to Review Comments**
* * *
**Response to the comments from the Editor**

**Comments:** I received review comments from two reviewers, both confirm the contribution of the manuscript, and think it's could be a good paper and also would be interesting to the HESS readers. Both reviewers think the manuscript need revision, and provide specific suggestions to revise the manuscript.

After receiving the comments, I carefully read the manuscript again, and concur with the two reviewers. The manuscript may need some rearrangement as suggested by the second reviewer, and the writing need to be polished.

**Response**

Accepted. The comments and critiques as noted by the editor and reviewers have been fully addressed in the revised manuscript. Our specific responses to these noted comments and critiques are provided in a point-by-point reply given below. The English writing was thoroughly revised by ourselves and an English copyediting service.
* * *
**Response to Reviewer #1's Comments**

**General Comments:** This study addresses an important issue of hydrological connectivity between glaciers in high mountains and river in the low plain within the alpine headwater catchment with big elevation difference and complex hydrogeological settings. The hydraulic head, temperature, and chemical and isotopic composition of groundwater, streamflow, precipitation and glacier meltwater were monitored along altitude gradient. The work has produced a remarkably rich data set that is clearly presented by the authors. The authors interpret the data to indicate that supra- and subpermafrost aquifers, as well as stream channels and slope surfaces, play an important role in transporting glacier, snow-meltwater and precipitation from the high mountains to the plain and then to the mainstem. The authors also suggest that a decline in hydro logical connectivity between the piedmont plain aquifer and the downstream channel in cold seasons may be the mechanism maintaining streamflow

(baseflow) in winter. It is worth pointing out that the authors present a logical and clearly illustrated conceptual model of hydrological connectivity in the alpine catchment by combining the above results. Given the wide distribution of this kind of headwater "mountain-plainriver" catchments in the Qinghai-Tibet Plateau and other cold regions, this conceptual model may contribute fundamentally to permafrost hydrology and can be more broadly utilized. The authors tentatively suggest that river icing and riverbank soil freezing may form a confining layer to reduce groundwater discharge from the plain to the stream, i.e., reduce the hydrological connectivity between the two pools. This is a very interesting hypothesis that can expand the existing mode for interpreting the slow release of stored groundwater during cold seasons, and it may be testable using field hydrometric measurement and numerical simulation. Overall the manuscript is well written and quite clear. I also have a few minor comments that I hope the authors to address before publication as listed in below.

**Response:** We thank the reviewer for carefully evaluating our manuscript and for constructive and helpful comments and suggestions. Our specific responses are given in below.

**Comment 1:** P2, L5: 'surface-water' should be 'surface water'.

**Response:** Changes have been made as suggested.

**Comment 2:** P2, L24 and L25: Two ';'s after 'hydrogeological' should be type errors.

**Response:** The type error has been corrected as suggested.

**Comment 3:** P3, L6: 'Heihe Basin' should be 'Heihe River Basin'.

**Response:** 'Heihe Basin' has been changed to 'Heihe River Basin'.

**Comment 4:** P3, L22: 'Qinghai-Tibet plateau' should be 'Qinghai-Tibet Plateau'.

**Response:** 'Qinghai-Tibet plateau' was changed to 'Qinghai-Tibet Plateau'.

**Comment 5:** P4, L15: Is 'the October to May cold season' a type error? 'ice covered' should be 'ice-covered'.

**Response:** Here we mean that the period from October to May is cold season. To avoid confusion, we have changed 'the October to May cold season' to 'the cold season (from October to May)'. 'ice covered' has also been changed to 'ice-covered' as suggested.

**Comment 6:** P6, L14: Citation is missing for the Gran titration method.

**Response:** We have added the citation for the Gran titration method as given in below.

*Gran G.: Determination of the equivalent point in potentiometric titrations. Part II, Analyst, 77: 661-671, 1952.*

**Comment 7:** P7, L27: What value does the $\delta 13C_{rech}$ take?

**Response:** As described in the texts (P8, L14-15 in the revised manuscript), the $\delta^{13}C_{rech}$ was taken $-18‰$ as suggested by Han et al. (2011) for north China.

**Comment 8:** P8, L22-23: This sentence is hard to understand. Please rewrite it.

**Response:** We have rewrote this sentence as following to make it more readable. The revised sentence reads now:

*"Although the groundwater depth differed greatly between the cold and warm seasons, it was relatively stable during each of the two seasons."*

**Comment 9:** P10, L15: Two 'respectively' should be removed.

**Response:** They have been deleted as suggested.

**Comment 10:** P11, L9-17: These results contrast with the statements in Abstract section.

**Response:** We have revised the abstract and make them to be consistent. In the revised abstract, we deleted the sentence "$^3$H and $^{14}$C data indicated that the age of supra- and sub-permafrost groundwater, and groundwater in Quaternary aquifer of seasonal frost zone, ranges from 30-60 years."

**Comment 11:** P13, L9: 'in water' should be 'in water table'.

**Response:** Change has been made as suggested.

**Comment 12:** P13, L29: I don't think that the dry sediment layer at depths between 12 m and 12.5 m is related to the subpermafrost groundwater.

**Response:** This sentence has been deleted.

**Comment 13:** P13, L29-31: Citation is missing for this statement.

**Response:** We have added citation for this statement as shown in below.

*Zhang, R., Liang, X., Jin, M., Wan, L., Yu, Q.: Fundamentals of Hydrogeology (6$^{th}$ Edition) (in Chinese), Geological Publishing House, Beijing, 2011.*

**Comment 14:** P14, L24-25: Citation is missing for this statement.

**Response:** We have added citation for this statement as shown in below.

*Clark, I. D. and Fritz, P.: Environmental Isotopes in Hydrogeology, CRC Press/Lewis Publishers, Boca Raton, Florida, USA, 1997.*
* * *
**Response to Reviewer #2's Comments**

**General comment:**

The authors studied the role of permafrost in controlling groundwater flow and the hydrological connections between glaciers in high mountain and river in the low plain

with hydraulic head, temperature, geochemical, and isotopic data. The paper is generally well written, and should be of very interest to the research community.

**Response:** We thank the reviewer for carefully evaluating our manuscript and for constructive and helpful comments and suggestions. Our responses to the specific comments are provided in a point-by-point reply given below.

**Comment 1:** Legend of Fig.12 should be explained clearly, such as the status of runoff (groundwater, surface water) should be depicted.

**Response:** We have revised Fig. 12 (as shown below) to make it more clear. Since there are many symbols in the figure, it is somewhat inconvenient and unclear if explain them in figure caption. Thus, we added a legend in Fig.12 to explain the meaning of different symbols used in the figure.

[Figure]

**Comment 2:** The resolution and framework of Fig. 10 should be improved.

**Response:** The resolution of Fig.10 is already 600 dpi. However, for comparison, we used the same Y-axis scale for three sub-plots (b), (c) and (d). This is the main reason why five lines in sub-plot (c) are too close to be distinguished clearly. The sub-plot (c) would have been more clear and aesthetic if a smaller scale was used for Y-axis. However, given that this figure is designed to show the difference in spatio-temporal variations of $\delta^{18}O$ between three water pools (i.e., well water, spring and stream), and provide insights on their hydrological connections, this framework can yield more

valuable information compared to that with varying scales. For example, as
mentioned in the manuscript (P12, L24-25 in the revised manuscript), spring waters
showed the smallest variation in $\delta^{18}O$ among three water pools, indicating a weaker
linkage with surface water, and probably a larger recharge area or/and a longer
residence time (in well-mixed). However, we tried our best to revise this figure to
make it more clear as shown in below.

[Figure]

**Comment 3**: The conclusions need to be improved, the author should tell the most
important conclusion by the simple statement at this part.

**Response:** We have revised the conclusions to focus on the most important things. The revised conclusions are as following:

*"Groundwater studies in permafrost area are challenging because of the limited infrastructure and the short field season. These conditions favor the use of geochemical and isotopic tracers in baseflow and perennial springs to supplement hydrogeological data to elucidate recharge conditions and flow paths. By selecting a representative catchment in the headwater regions of the Heihe River, Qinghai-Tibet Plateau as study site, this research employed the groundwater head, temperature, geochemical, and isotopic information to determine the roles of groundwater in permafrost and seasonal frost zone for hydrologically connecting waters originating from glaciers in the high mountains to lower elevation rivers.*

*Our field measurements show the co-occurrence of supra-, intra- and subpermafrost groundwaters in the headwater regions of the Heihe River. To the best of our knowledge, this is the first report of the occurrence of sub- and intrapermafrost groundwaters in this region. The moraine and fluvio-glacial deposits on the planation surfaces of higher hills, which are commonly distributed in the headwater regions of the Heihe River, provide a major reservoir for the storage and flow of sub- and intrapermafrost groundwater. The subpermafrost groundwater on the planation surface was interconnected to the surface hydrological processes and recharged by suprapermafrost groundwater and glacier and snow meltwater. The results of this study could shed new lights on the understanding of the groundwater flow and its interaction with surface water at other catchment, as well as improve the evaluation and management of water resources in the headwater regions of the Heihe River.*

*Glacier and snow meltwater were transported from the high mountains to the plain through stream channels, slope surfaces, and supra- and subpermafrost aquifers. The groundwater in the piedmont plain within seasonal frost zone was mainly recharged by the lateral flow from the supra- and subpermafrost aquifers and the seepage of streams, and was discharged as baseflow into the Hulugou stream in the north gorge. A rapid transfer of groundwater from the south top to the north base of the plain occurred during the warm season, while the stored groundwater was slowly released during the cold season. This seasonal variation of the aquifer in water-conduction capacity was interpreted by two mechanisms: (1) surface drainage via the stream channel, analogous to the "fill and spill" mechanism in hillslope hydrology. The*

*narrowing of aquifer from the wide plain to the gorge led to a relatively high water table near the gorge, preventing it from dropping below the channel bed and maintaining a perennial flow in the downstream. This addresses the rapid transfer of groundwater from the top to the base of the plain and the stable water table in front of the gorge during the warm season; and (2) subsurface drainage to an ephemeral artesian aquifer confined by stream icing and seasonal frost. The stream icing and seasonal frost not only blocked the groundwater discharge, but also changed the bottom of the gorge into a confined aquifer during the cold season, leading to an increase in the downstream groundwater head and a decrease in the hydraulic gradient between the wide plain and the narrow gorge. The second mechanism elucidates the slow release of stored groundwater from the plain and the low baseflow in channel throughout the cold season."*

**Comment 4:** Page 11, the value of $\delta^2 H$ and $\delta^{18}O$ indicate that suprapermafrost groundwater had experienced strong evaporation, but the hydrogeochemistry also suggest the suprapermafrost groundwater has rapid flow. It should be explained more clearly.

**Response:** We have revised this part as suggested to express it more clearly. The revised statement is as following:

*"The low TDS, $Cl^-$ and $Na^-$ concentrations and the $HCO_3$-Ca water type suggest that suprapermafrost groundwater had experienced insufficient water-mineral interaction, probably caused by a relatively short residence time or flow path. This is further supported by the highest $^{14}C$ activity in the suprapermafrost groundwater among all samples (Table 2), which is 96.34 pmC, and by a 15.11 TU $^3H$ concentration which is close to the atmospheric value and an indicator of modern water (Zhai et al., 2013). Though occurrance on a relatively flat planation surface, the suprapermafrost groundwater was actually easy to drain because the planation surface adjoins the lower slopes in three directions. In addition, the suprapermafrost aquifer is fairly thin and rich in organic matter with high permeability. Therefore, the suprapermafrost groundwater may have a high renewal rate. The enriched $^2H$ and $^{18}O$ isotopes indicate that suprapermafrost groundwater had also experienced a certain degree of evaporation (Figure 8). These two conclusions are not contradictory to each other given the high local evaporation and shallow suprapermafrost groundwater depth. The shallow groundwater depth may also result in very short flowpaths for the majority of the waters and relatively short contact time for chemical reactions*

*between the water and the soils (Frey et al., 2007; Stotler et al., 2009; Vonk et al., 2015)."*

**Comment 5:** The English of the whole manuscript need to be improved.

**Response:** We have tried our best to edit the English and we have also asked the professional English editing service to polish the English writing.

[revised manuscript text omitted]

**4.6 Radioactive isotopes and groundwater age**

The $^3H$ concentrations were 15.11 TU in the groundwater from the 1.5m well at cluster WW04, between 16.20 and 24.18 TU in the groundwater at clusters WW01 and WW03, and between 13.61 and 43.59 TU in the springs in the sloping plain (Table 2). Except for one spring sample (QW05), the $^3H$ concentrations of all samples were < 30 TU, indicating that the groundwater was recharged by recent precipitation and some "bomb"

related [3]H is possibly presented  (Zhai et al., 2013). Along with flow path, the $\delta^{13}$C-$_{DIC}$ in groundwaters increased from the permafrost zone with values between -13.6 and -16.77‰ to the top of the sloping plain with values  ~ −-8.79‰, and further to the base of the sloping plain with values  ~ −-5.09‰ (Table 2). Opposite to $^{13}$C-$_{DIC}$ trend,  the $^{14}$C activity decreased  from permafrost zone  with values  between 35.51 and 96.34 pmC  to the top of the sloping plain with values ~51 pmC, and further to the base of the sloping plain with values ~ 44 pmC (Table 2).  $^{14}$C ~~values than values in groundwater and spring samples from the sloping plain, showing a general increasing trend from the permafrost zone to higher locations of the seasonal frost zone, and further to the lower elevation groundwaters. Except for the 24.3 m well within cluster WW04, the corrected $^{14}$C ages of all samples were negative, indicating that they were derived from modern precipitation (Clark and Fritz, 1997).samplefromwithin, with the $\delta^{13}$C of −16.77‰,~~ had a relatively old corrected $^{14}$C age of 1627 yr. The other groundwaters exhibited negative corrected $^{14}$C ages, indicating that they were derived from modern precipitation (Clark and Fritz, 1997).

**5. Discussion**

**5.1 Exchange and pathways of groundwater in the permafrost zone**

The groundwater in the 1.5 m well at cluster WW04 occurred within the active layer and thus was recognized as suprapermafrost groundwater, which was previously reported in the study area (Cao, 1977). Within the permafrost layer with a thickness of 20 m (2−22 m below ground), the groundwater was found in a talik at the depths between 12 and 12.5 m (in the 12 m well). It was considered as intrapermafrost groundwater. The underlying subpermafrost groundwater in the 24.3 m well was observed in the field, which was further evidenced by the slightly increased temperature and the distinct hydrogeochemistry. The intra- and subpermafrost groundwater had not been reported before this study.

**5.1.1 Suprapermafrost groundwater**

~~The low TDS, Cl⁻ and Na⁻ concentrations and the HCO₃-Ca water type suggest that suprapermafrost groundwater had experienced insufficient water-mineral interaction, probably caused by a relatively short residence time. This is supported by the highest $^{14}$C activity in the suprapermafrost groundwater among all samples (Table 2), which is 96.34 pmC and close to the atmospheric value (Clark and Fritz, 1997), and a 15.11 TU $^3$H concentration, which is an indicator of modern water (Zhai et al., 2013). Though occurring on a relatively flat planation surface, the suprapermafrost groundwater is actually easy to drain because the planation surface adjoins the lower slopes in three directions (see below). Add to that the fact that suprapermafrost aquifer is fairly thin whereas rich in organic matter with high permeability, one can understand why it may have a high renewal rate. However, the enriched $^2$H and $^{18}$O isotopes, along with samples' position relative to the LMWL in~~

~~the $\delta^2$H vs. $\delta^{18}$O plot, indicate that suprapermafrost groundwater had also experienced a certain degree of evaporation (Figure 8). These two conclusions are not in conflict when considering the high local evaporation (376–650 mm/yr) and shallow suprapermafrost groundwater depth (0–1.5 m below ground). The high groundwater table may also result in very shallow flowpaths for the majority of the water and few possibilities for chemical reactions between the discharging water and the deep mineral soil (
[revised manuscript text omitted]